# Presynaptic contact and activity opposingly regulate postsynaptic dendrite outgrowth

**Emily L Heckman, Chris Q Doe***

Institute of Neuroscience, Howard Hughes Medical Institute, University of Oregon, Eugene, United States

**Abstract** The organization of neural circuits determines nervous system function. Variability can arise during neural circuit development (e.g. neurite morphology, axon/dendrite position). To ensure robust nervous system function, mechanisms must exist to accommodate variation in neurite positioning during circuit formation. Previously, we developed a model system in the *Drosophila* ventral nerve cord to conditionally induce positional variability of a proprioceptive sensory axon terminal, and used this model to show that when we altered the presynaptic position of the sensory neuron, its major postsynaptic interneuron partner modified its dendritic arbor to match the presynaptic contact, resulting in functional synaptic input (Sales et al., 2019). Here, we investigate the cellular mechanisms by which the interneuron dendrites detect and match variation in presynaptic partner location and input strength. We manipulate the presynaptic sensory neuron by (a) ablation; (b) silencing or activation; or (c) altering its location in the neuropil. From these experiments we conclude that there are two opposing mechanisms used to establish functional connectivity in the face of presynaptic variability: presynaptic contact stimulates dendrite outgrowth locally, whereas presynaptic activity inhibits postsynaptic dendrite outgrowth globally. These mechanisms are only active during an early larval critical period for structural plasticity. Collectively, our data provide new insights into dendrite development, identifying mechanisms that allow dendrites to flexibly respond to developmental variability in presynaptic location and input strength.

**\*For correspondence:**
cdoe@uoregon.edu

## Editor's evaluation

The findings reported in this article build nicely on previous work regarding the specification and growth of post-synaptic dendrites. Here the authors conduct an elegant genetic and anatomical analysis defining two opposing mechanisms that regulate post-synaptic dendrite morphogenesis and/or stabilization: presynaptic contact and neuronal activity. The data are of uniformly high quality, support the authors' major conclusions and offer important new insights into this fundamental aspect of circuit development.

## Introduction

Neural circuit organization dictates circuit function, influencing behavior, cognition, and perception. While developmental programs in genetically identical animals produce similar final products, variability arises during circuit wiring to produce differences in cellular morphology, synaptic partnerships, and numbers of synapses between partners (*Mohr et al., 2004*; *Chou et al., 2010*; *Caron et al., 2013*; *Linneweber et al., 2020*; *Churgin et al., 2021*; *Courgeon and Desplan, 2019*; *Couton et al., 2015*; *Goodman, 1978*; *Tobin et al., 2017*; *Witvliet et al., 2021*). Such variability can arise innately due to stochastic processes (e.g. filopodial extension/retraction, lateral signaling, gene expression)

(*Özel et al., 2015*; *Troemel et al., 1999*; *Wernet et al., 2006*) or when environmental factors impinge on development (e.g. rearing temperature, sensory experience) (*Hubel et al., 1977*; *Kiral et al., 2021*; *Shatz and Stryker, 1978*). Neural development must be flexible to account for these variations to generate robust circuit function.

To ensure robust circuit function, developing neurons must exhibit specificity and flexibility – specificity in synaptic partner choice and flexibility to respond to variability in partner neuropil territory. If the location and strength of synaptic inputs can vary, how do surrounding neurons adapt such that they receive the right type and right amount of input? Dendrites are the major sites of synaptic input onto a neuron. Studies have shown that dendrites alter their morphology in response to varying levels of synaptic input (*Ackerman et al., 2021*; *Takeo et al., 2021*; *Tripodi et al., 2008*), and more recently we showed that dendrites can alter their position when a presynaptic partner is routed to an alternate neuropil location (*Valdes-Aleman et al., 2021*). Dendrites are clearly capable of structural plasticity, yet how they appropriately respond to a variable developmental landscape is unclear.

Here, we investigate how and when developing dendrites accommodate wiring variation to ensure robust circuit connectivity. The goal of our study was to determine the cellular mechanisms used by dendrites to compensate for variability in presynaptic axon placement and input strength to facilitate functional connectivity. To study these mechanisms in vivo, we used a model system consisting of the *Drosophila* larval dbd sensory neurons and their postsynaptic partners, the A08a interneurons. dbd neurons are proprioceptors and sense body-wall segment elongation during peristalsis (*Suslak et al., 2015*; *Vaadia et al., 2019*); their activity is required for maintaining efficient crawling speed (*Hughes and Thomas, 2007*). dbd neurons form segmentally repeated connections with A08a in the larval abdominal segments. A08a has two distinct dendritic domains: lateral and medial. All major inputs to the A08a dendritic domain synapse with a single dendrite, either lateral or medial; the dbd sensory neuron synapses with the medial dendrite (*Sales et al., 2019*; *Schneider-Mizell et al., 2016*). The function of A08a neurons in larval behavior has yet to be determined, although their wave-like activity during locomotion suggests that A08as are involved in regulating larval peristalsis (*Itakura et al., 2015*; *Sales et al., 2019*).

Leveraging strategies to manipulate presynaptic activity levels and presynaptic contact with postsynaptic dendrites, we find that there are two opposing mechanisms used to regulate robust partner matching between dbd and A08a: presynaptic contact promotes dendrite outgrowth locally, while presynaptic activity inhibits elongation of postsynaptic dendrites globally. These two strategies highlight the important role of presynaptic inputs in regulating the placement and subsequent elaboration of postsynaptic dendrites.

## Results
### dbd-Gal4 labels the dbd sensory neuron prior to formation of presynaptic contacts

We sought to induce variability in the placement of the dbd axon to determine the extent to which its connectivity with A08a is stringently required at the medial dendrite. To do this we first needed genetic access to dbd prior to its interaction with A08a. We previously used *165*-Gal4 (subsequently referred to as *dbd-Gal4*) to label the dbd sensory neuron (*Sales et al., 2019*; *Valdes-Aleman et al., 2021*), but the onset of dbd-Gal4 expression was not determined. Here, we use dbd-Gal4 to drive expression of a myr::HA tag, and the 22C10 antibody to label all sensory neurons as a landmark. We observed the first expression of dbd-Gal4 in dbd neurons at embryonic stage 14. At this time, the axons have just entered the dorsal central nervous system (CNS) as immature growth cones (*Figure 1A*), and would not yet have contacted the more ventral neuropil domain occupied by the A08a dendrites. By stage 15, the dbd neurons formed anterior/posterior bilateral branches adjacent to the midline of the neuropil (*Figure 1B*), followed by further elaboration to link adjacent segments in stage 17 (*Figure 1C*). By this time, dbd axon terminals occupy a more ventral region of the neuropil where A08a medial dendrites are formed (*Schrader and Merritt, 2000*; *Zlatic et al., 2003*). These patterns of dbd projection into the CNS are schematized in *Figure 1F*.

The A08a interneuron is labeled using a previously characterized LexA (*26F05-LexA*) to express *LexAop-myr::V5* in A08a and its dendritic arbors. This line first labels A08a in the early hours of the first larval instar (*Figure 1D and F*). By this time, A08a has contacted dbd and has developed its

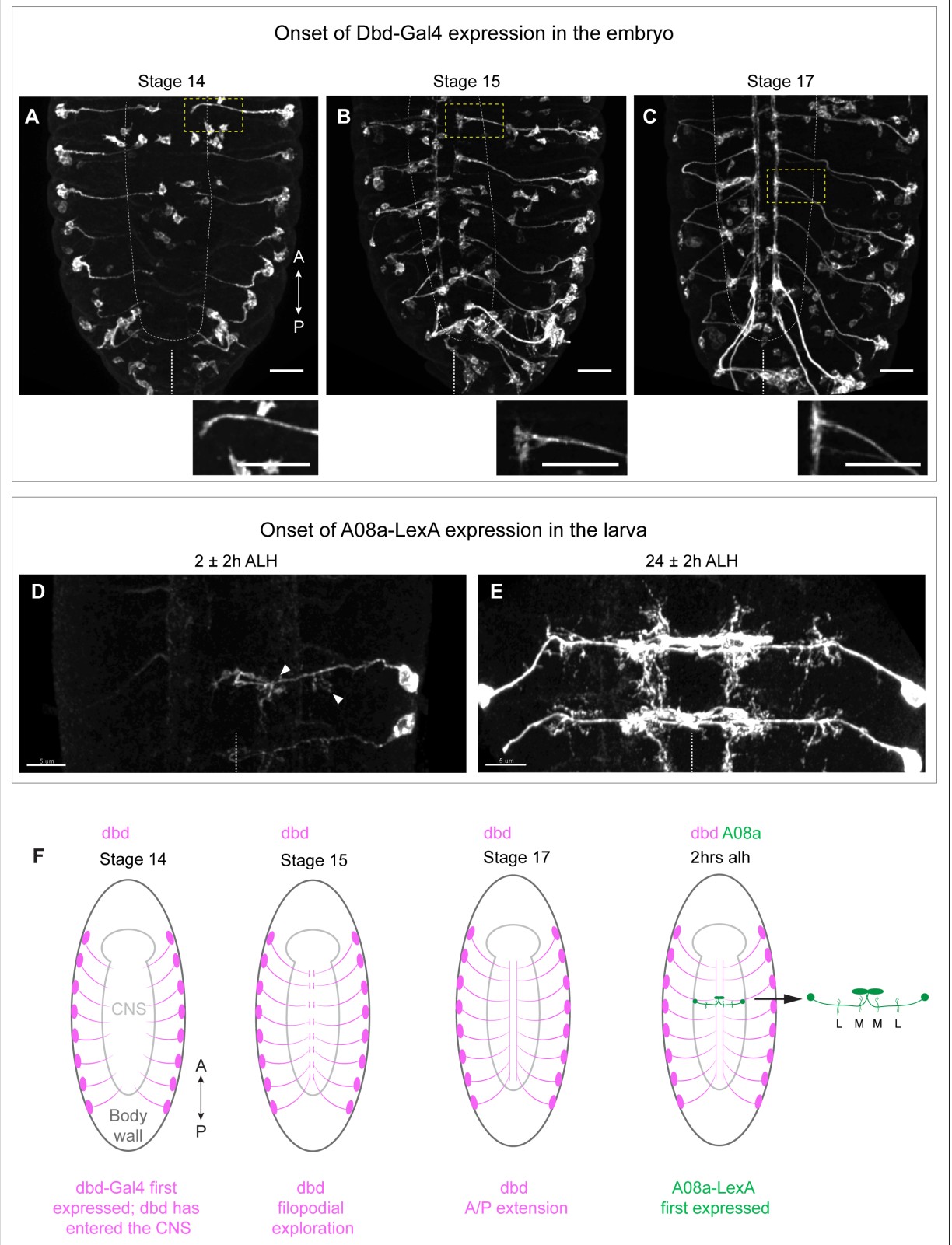

**Figure 1.** Onset of dbd-Gal4 and A08a-LexA expression in the embryo and early larva. (**A–C**) Stage 14, 15, and 17 fixed embryos. dbd-Gal4 pattern labeled with smGdP-myr::HA (white). Stage 14, n=4 animals; stage 15, n=13 animals; stage 17, n=17 animals. Cell bodies not in the body wall are likely part of the gut. Insets show zoomed in view of dbd outlined by yellow dashed box. White dashed line indicates outline of central nervous system (CNS). Midline indicated by white dashed line at the bottom of each image. Scale bars, 20 μm. (**D**) Dorsal view of VNC at 2±2 hr after larval hatching (alh)

*Figure 1 continued on next page*

Figure 1 continued

showing A08a-LexA expression pattern labeled with smGdP-myr::V5 in white. Arrow heads indicate the medial and lateral dendrites. In this image, LexA expression is on in A1R, weakly in A2R, and not at all in the opposing A1L and A2L hemisegments. n=7 animals. (**E**) Dorsal view of VNC at 26±2 hr alh. A08a-LexA is expressed robustly in all hemisegments at this time. n=7 animals. Images are max intensity projections. Scale bars, 5 µm. Midline indicated by white dotted line. (**F**) Illustrations summarizing results in A–D.

The online version of this article includes the following figure supplement(s) for figure 1:

**Figure supplement 1.** Neurons with the most synapses with A08a.

characteristic lateral and medial dendritic arbors. The timing of A08a-LexA expression precludes us from knowing the state of A08a dendrite development at the time of first contact with dbd. However, we conclude that dbd-Gal4 labels the dbd sensory neuron prior to its growth into the A08a neuropil domain, and thus it is an appropriate tool for manipulating dbd prior to the establishment of dbd-A08a synaptic connectivity.

## The dbd sensory neuron locally promotes dendrite elongation in the A08a interneuron

In a previous study, we tested the ability of dbd and A08a to compensate for developmentally induced wiring variation (*Sales et al., 2019*). We tested for stringent specificity in dbd connectivity with the medial arbor by using genetic methods to target the dbd axon to the lateral neuropil. The dbd axon terminal could be misrouted to the intermediate and lateral neuropils through misexpression of repulsive axon guidance receptors, Robo-2 and Unc-5. dbd could form synapses at both intermediate and lateral dendritic arbors, and the strength of functional connectivity when dbd synapsed with the lateral arbor was indistinguishable from wild-type dbd-A08a connectivity (*Sales et al., 2019*).

Surprisingly, when dbd was targeted to lateral or intermediate neuropil domains, A08a produced an ectopic dendritic arbor to match the presynaptic contact (*Valdes-Aleman et al., 2021*). Here, we extended and confirmed these findings by showing that the cumulative distribution of A08a dendrite volume corresponds to the location of dbd input (*Figure 2A–D*). The increase in dendrite arbor volume was most striking in the intermediate zone of the A08a dendritic domain where there are few arbors present in wild type (*Figure 2A–B*). Interestingly, when dbd was targeted to intermediate or lateral neuropil regions, the volume of the medial dendrite was decreased (*Figure 2D*).

The rearrangement of dendrite volume could be due to novel dendrite outgrowth from the main A08a neurite, or due to elaboration of pre-existing lateral or medial arbors. To distinguish between these possibilities, we plotted the frequency of branch points off the main A08a neurite across the lateral-medial axis. We found that when dbd is targeted to the intermediate zone, there was an increase in the frequency of dendritic branches off the primary A08a neurite in the intermediate zone, supporting the idea that the observed increase in dendrite volume in the intermediate zone is due to novel dbd-promoted dendrite outgrowth (*Figure 2E–F*). We conclude that presynaptic contact can locally promote postsynaptic dendrite outgrowth, ultimately ensuring robust partner matching.

## dbd ablation results in A08a lateral dendrite expansion

Our findings led us to investigate the mechanisms of ectopic A08a dendrite establishment. To test the hypothesis that dbd contact promotes local dendrite outgrowth, we genetically ablated dbd using the *dbd-Gal4* line driving expression of the pro-apoptotic gene *hid*. If dbd promotes medial dendrite outgrowth, we would expect dbd ablation to reduce the A08a medial dendritic arbor.

We confirmed dbd ablation by 22C10 staining, which labels all sensory neurons. Embryonic and larval sensory neuron cell bodies are located in the body wall, and have stereotyped positions. 22C10 marks the dbd neuron cell body, positioned at the base of the dorsal-most cluster of sensory neurons (*Figure 3—figure supplement 1A and C*; *Ghysen et al., 1986*). In addition, we also used the absence of dbd-Gal4-driven myr::HA to identify segments where dbd was ablated (*Figure 3C*; *Figure 3—figure supplement 1D*; Figure 6C). Hid expression indeed led to a loss of dbd neurons, as detected through the absence of both 22C10[+] and HA[+] cell bodies from the body wall (*Figure 3—figure supplement 1B and D*). After confirming that Hid expression eliminated dbd, we next assayed control and dbd ablation larvae at 24±2 hr after larval hatching (alh) for A08a dendrite length. A08a dendrites were reconstructed using the Imaris software Filaments tool (*Figure 3B' and D'*). As expected, controls

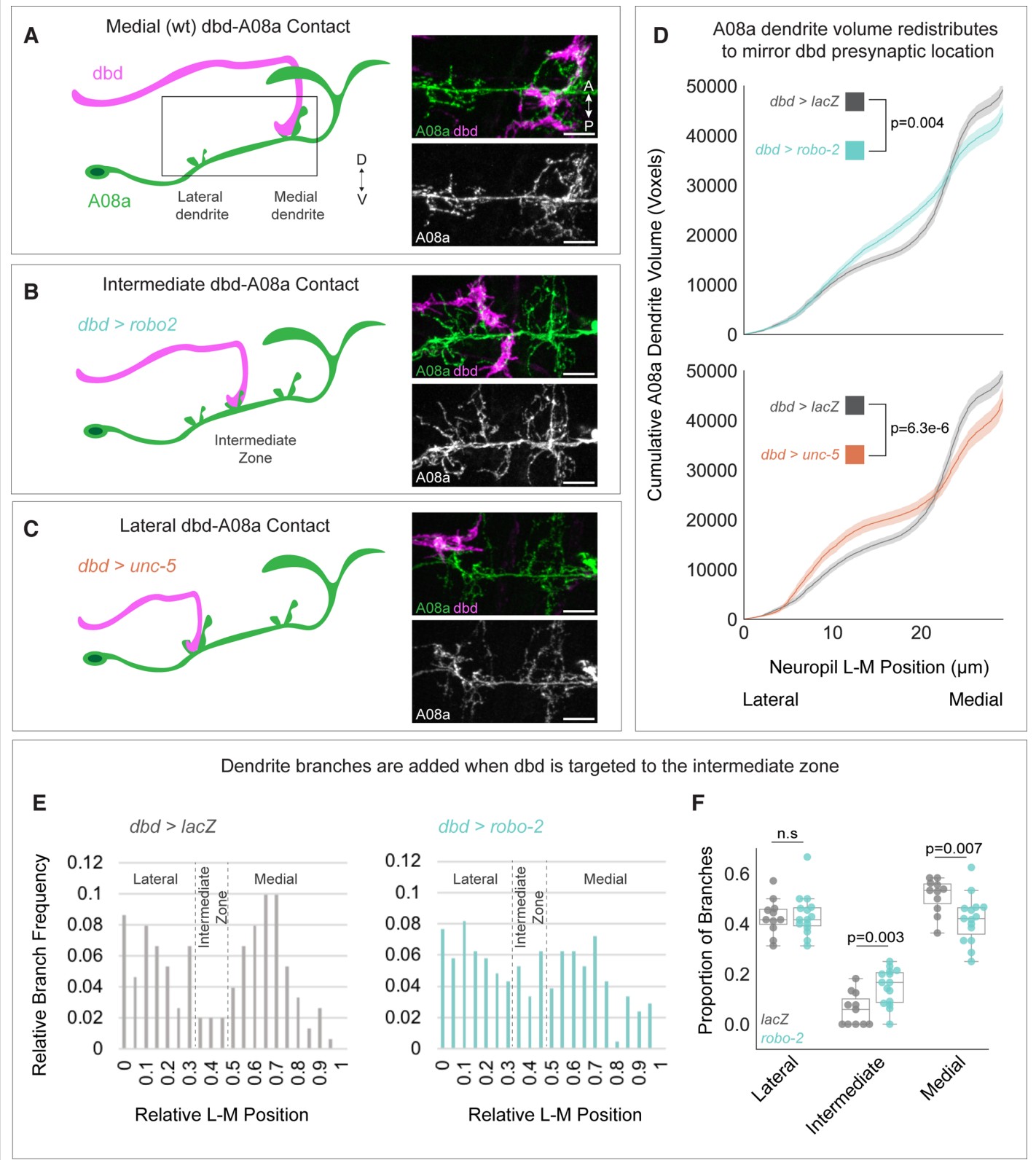

**Figure 2.** Dendrite development is promoted by presynaptic axons. (**A**) Left: Illustration of wild-type dbd (pink) and A08a (green). dbd projects to the A08a medial dendrite. Right: A08a dendritic domain (boxed region shown in cartoon). dbd (pink) contacts the medial A08a dendrite (green). Secondary image of A08a channel alone (white) shows two distinct dendritic domains. (**B**) Left: Robo2 misexpression leads dbd to project to the A08a intermediate dendritic domain. Right: dbd contacts the intermediate A08a dendritic domain, where there are ectopic dendrites. (**C**) Left: Unc-5 misexpression leads

*Figure 2 continued on next page*

*Figure 2 continued*

dbd to project to the A08a lateral dendrite. Right: dbd contacts the lateral A08a dendrite, where there is ectopic dendritic material. Scale bar, 5 µm. Micrographs are from larvae aged 24±4 hr after larval hatching (alh). (**D**) Cumulative distribution of A08a dendrite volume (voxels) across the lateral-medial axis in conditions where dbd projects to the medial (gray, n=17 cells, 11 animals), intermediate (cyan, n=20 cells, 10 animals), or lateral (orange, n=11 cells, 10 animals) A08a dendrite. Solid line = mean distribution; shaded area = standard error of the mean (SEM). A08a dendrites receiving input from intermediate or lateral dbd neurons have significantly different volume distributions from wild-type dendrites (lacZ vs. robo-2, p=0.004; lacZ vs unc-5, p=6.3e-6. Two-way Kolmogorov-Smirnov test). Note that the control LacZ trace is the same for top and bottom panels. (**E**) Relative lateral-medial distribution of A08a dendrite branch points from the main A08a neurite in conditions where dbd projects to the medial (gray, n=11 cells from 7 animals) or intermediate (cyan, n=15 cells from 8 animals) dendritic domain. Lateral, intermediate, and medial boundaries are demarcated based on the local minimum of the LacZ distribution. (**F**) Proportion of branches occupying lateral, intermediate, and medial A08a dendritic domains when dbd projects to the medial dendrite (gray, n=11 cells from 7 animals) or intermediate zone (cyan, n=15 cells from 8 animals). Individual points represent single cells. When dbd projects to the intermediate domain, there are more A08a branches in the intermediate domain and fewer in the medial domain (Lateral Domain: lacZ vs. robo-2, p=0.85; Intermediate Domain: lacZ vs. robo-2, p=0.003; Medial Domain: lacZ vs. robo-2, p=0.007. Statistics computed using two-tailed unpaired t-test with unequal variance).

showed well-branched medial and lateral dendritic arbors by immunostaining (*Figure 3A–B*) and in the Imaris reconstructions (*Figure 3B'*). In contrast, dbd ablation led to qualitatively enlarged lateral dendrites by immunostaining (*Figure 3C–D*) and in the Imaris reconstructions (*Figure 3D'*). Quantification confirmed that complete dbd ablation led to A08a lateral dendrites that were longer and more branched (*Figure 3E–F*). In contrast, there was no significant change in medial dendrite length or branching when dbd was ablated (*Figure 3E–F*).

Hid overexpression resulted in variable numbers of ablated dbd neurons across samples (*Figure 3—figure supplement 1B*; Figure 6B). We could therefore test whether there is a correlation between the number of dbd neurons innervating a segment and the total A08a dendrite length. In wild type, there are 4 dbd neurons innervating a single VNC segment, two per hemisegment. We found that when 1–2 of the 4 dbd neurons are ablated, A08a dendrite length is not significantly different than in controls. However, when 3–4 of the 4 dbd neurons are ablated, A08a lateral dendrite length is significantly increased (*Figure 3G*). This finding suggests that a threshold level rather than a linear summation of dbd input stabilizes A08a dendrite outgrowth.

The expansion of lateral A08a dendrite length was surprising, as dbd does not contact the lateral dendrite in wild-type animals (*Sales et al., 2019*; *Schneider-Mizell et al., 2016*). The growth in lateral dendrites following dbd ablation revealed that dbd provides negative regulation of A08a dendrite outgrowth, in addition to the positive mechanism revealed by the dbd lateralization experiments. Moreover, the negative mechanism acts throughout the neuron, rather than locally as does the positive mechanism. A likely source of a neuron-wide inhibitor of dendrite outgrowth is neuronal activity, which negatively regulates dendrite arbor size in multiple systems (*Ackerman et al., 2021*; *Shen et al., 2020*; *Tripodi et al., 2008*; *Wu and Cline, 1998*). We address this possibility in the next section.

## dbd activity globally inhibits A08a dendrite outgrowth

Based on the L1 larval TEM connectome, dbd and A08a neurons are among each other's strongest synaptic partners. For dbd, the top 10 synaptically connected 'downstream' neurons are shown in *Table 1*, with A08a ranking #6; conversely, for A08a, the top 10 input neurons are shown in *Table 2*, with dbd being the #1 input to A08a (*Tables 1 and 2*; *Figure 1—figure supplement 1*). Based on synapse number and neurotransmitter identity, dbd is the major source of A08a excitation. Taken together with our results from the previous section, we hypothesized that inhibiting dbd excitatory neurotransmission would reduce A08a dendrite length.

To test if dbd activity negatively regulates A08a dendrite size, we used two methods to silence dbd synaptic activity throughout development. The first was dbd-specific expression of either tetanus toxin light chain (TNT) or mutationally inactive TNT as a negative control. TNT cleaves the synaptic vesicle protein synaptobrevin, inhibiting evoked synaptic vesicle release (*Sweeney et al., 1995*). A previous study showed that dbd silencing led to slowed and uncoordinated larval locomotion (*Hughes and Thomas, 2007*). We validated that our silencing tools were effectively inhibiting dbd activity by comparing larval crawling behavior between control and dbd-silenced animals. Dbd-silenced animals had fewer waves of crawling activity, consistent with the previous results (*Figure 4A–B and E*). Thus, we proceeded to ask if dbd silencing leads to expanded A08a dendrites. We found that constitutive

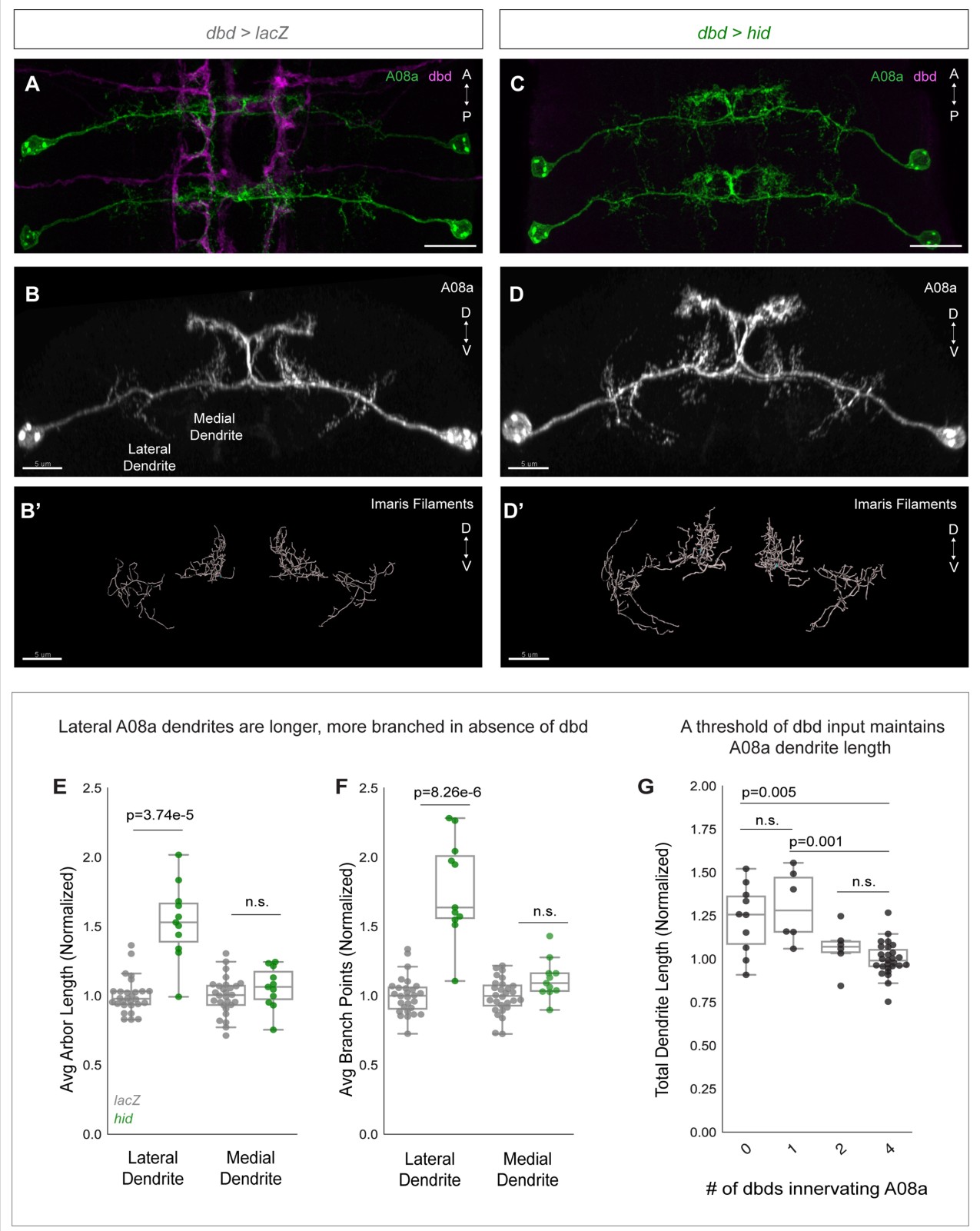

**Figure 3.** dbd ablation causes A08a lateral dendrite expansion. (**A**) Control VNC at 24±2 hr after larval hatching (alh), dorsal view. A08a neurons (green) are innervated by dbd neurons (magenta). Scale bar, 10 μm. (**B**) Control A08a neurons from abdominal segment 1 of (**A**), posterior view. Scale bar, 5 μm. (**B'**) Imaris filament reconstruction of A08a dendrites in (**B**). Scale bar, 5 μm. (**C**) dbd ablation VNC at 24±2 hr alh, dorsal view. A08a neurons (green) lack innervation from dbd neurons (magenta). (**D**) A08a neurons in dbd ablation background from abdominal segment 1 of (**C**), posterior view. Scale bar,

*Figure 3 continued on next page*

*Figure 3 continued*

5 µm. (**D'**) Imaris filament reconstruction of A08a dendrites in (**D**). Scale bar, 5 µm. (**E**) Average dendrite length of lateral and medial A08a dendritic arbors in LacZ (control, gray, n=28 animals) or Hid-expressing animals (green, n=11 animals). Hid animals have longer lateral dendrites (p=3.74e-5) but unchanged medial dendrite length (p=0.56). Values normalized to control mean for either lateral or medial arbor. Circles represent single-animal averages between left and right hemisegments. Values for Hid-expressing animals with 0 remaining dbd neurons innervating A1 segment are shown in (**E**) and (**F**). (**F**) Number of dendrite branch points of lateral and medial A08a dendritic arbors in LacZ (control, gray, n=28 animals) or Hid-expressing animals (green, n=11 animals). Hid animals have more branched lateral dendrites (p=8.26e-6) and unchanged levels of branching among medial dendrites (p=0.06). (**G**) Total A08a dendrite length when A08a is innervated by 4 (LacZ control, n=28 animals), 2 (Hid, n=6 animals), 1 (Hid, n=6 animals), or 0 (Hid, n=10 animals) dbd neurons. No statistical difference between 4 vs. 2 dbds (p=0.31) or 0 vs 1 dbd (p=0.50). 1 dbd vs. 4 dbds results in longer A08a dendrites (p=0.001) and 0 vs. 4 dbds results in longer A08a dendrites (p=0.005). Circles represent single-animal dendrite length summed across left and right hemisegments. Data are normalized to dendrites innervated by 4 dbd neurons (LacZ control). Statistics computed using two-tailed unpaired t-test with unequal variance. n.s.=not significant, p>0.05.

The online version of this article includes the following figure supplement(s) for figure 3:

**Figure supplement 1.** Validation of dbd ablation.

dbd silencing using TNT indeed resulted in longer, more branched lateral and medial A08a dendrites at 26±2 hr alh (*Figure 4F–I*).

We also used Shibire[ts] to chronically and specifically silence dbd (*Kitamoto, 2001*). Shibire[ts] animals were reared constitutively at 30°C and compared to temperature-matched negative controls expressing LacZ, as developmental temperature was recently shown to impact the extent of neurite branching and synapse formation (*Kiral et al., 2021*). Silencing of dbd with Shibire[ts] also impaired larval crawling efficiency (*Figure 4C–E*), and resulted in longer, more branched A08a dendrites (*Figure 4J–M*). Presynaptic activity from dbd is therefore necessary to prevent excessive postsynaptic dendrite outgrowth in A08a.

If presynaptic activity inhibits A08a dendrite outgrowth, we predicted that elevated levels of dbd activity would conversely result in shorter A08a dendrites. To activate dbd, we expressed the light-sensitive channelrhodopsin CsChrimson using dbd-Gal4 (*Klapoetke et al., 2014*). Animals were exposed to broad spectrum light throughout development and A08a dendrite length and complexity were assayed at 25±2 hr alh. Like our silencing experiments, we first confirmed CsChrimson function by assaying larval crawling efficiency. We found that CsChrimson-expressing larvae initiated fewer locomotor waves of activity (*Figure 5A–C*), confirming a previous report that activation of sensory neurons, including dbd, results in slowed larval locomotion (*Pulver et al., 2009*). After validating CsChrimson function, A08a dendrite length and complexity were assessed. Compared to negative controls expressing LacZ, dbd optogenetic activation led to a decrease in A08a dendrite length and branching. This effect was most pronounced at the medial dendrite; although some lateral arbors exhibited decreased length and branching, it did not reach statistical significance (*Figure 5D–G*). We present some possible reasons for this in the Discussion. The decreases in arbor length and branching were likely not due to excitotoxicity, as this method has been previously published in larval motor neurons without inducing excitotoxicity (*Ackerman et al., 2021*). We conclude that presynaptic activity is necessary and sufficient to restrict A08a dendrite outgrowth.

## A08a dendrite plasticity is confined to a critical period of development

Neurons in many animals exhibit transient structural plasticity in early developmental windows

**Table 1.** Top postsynaptic partners of dbd. Top 10 neurons with the most synapses with dbd in segment A1. Neurons must be observed in both left and right hemisegments and have a minimum of ≥3 synapses with dbd combined.

| Neuron identifier | Cell body location | Synapses with dbd in A1 |
|---|---|---|
| Clamp-3 | A2 | 42 |
| A02a | A1+A2 | 37 |
| Jaam-4 | A1 | 36 |
| JaamXX | A1 | 33 |
| Jaam-3 | A1 | 29 |
| A08a | A1 | 26 |
| A06 | A1 | 24 |
| A08s1 | A2 | 22 |
| A02d | A1 | 21 |
| A08s2 | A2 | 19 |
| A10x | A1 | 17 |
| Jaam-1 | A1 | 16 |

**Table 2.** Top presynaptic partners of A08a.

Top 10 neurons with the most synapses with A08a in segment A1. Neurons must be observed in both left and right hemisegments and have a minimum of ≥3 synapses with A08a combined.

| Neuron identifier | Cell body location | Synapses with A08a in A1 | A08a domain innervated | Neurotransmitter (NT) identity | NT reference |
|---|---|---|---|---|---|
| dbd | A1+A2 | 33 | Medial dendrites | Cholinergic | *Salvaterra and Kitamoto, 2001* |
| A02d | A1 | 30 | Medial dendrites | Glutamatergic* | *Kohsaka et al., 2014* |
| T11v? | T2 | 24 | Axon/output | | |
| A14a1 | A1 | 21 | Axon/output | | |
| A18g | A2+A3 | 19 | Axon/output | Cholinergic | *Hiramoto et al., 2021* |
| A02l | A1 | 16 | Lateral dendrites | Glutamatergic* | *Kohsaka et al., 2014* |
| A31x | A1 | 12 | Lateral dendrites | GABAergic | *Fushiki et al., 2016* |
| vbd | A1 | 12 | Lateral dendrites | Cholinergic | *Salvaterra and Kitamoto, 2001* |
| A14 dude | A2 | 11 | Axon/output | | |
| Drunken-6 | A1 | 7 | Lateral dendrites | | |

*Note that glutamate can act as an inhibitory neurotransmitter in the *Drosophila* CNS (*Liu and Wilson, 2013*; *Rohrbough and Broadie, 2002*).

that are termed 'critical periods' for plasticity (*Ackerman et al., 2021*; *Jarecki and Keshishian, 1995*; *LeVay et al., 1980*; *McLaughlin et al., 2003*). In *Drosophila* larvae, motor neuron dendrites remain plastic until 8 hr alh, after which astrocytes prevent further dendritic remodeling, at least through 22 hr alh (*Ackerman et al., 2021*). Larvae also undergo continuous neuronal arbor growth to scale with their increasing body size, which may require some neurons to remain adaptable to a changing cellular environment (*Gerhard et al., 2017*). We therefore wanted to test whether A08a dendrites remain plastic in later stages of larval life, or if they are subject to the same critical period as motor neurons. To do so, dbd neurons were conditionally ablated by expressing Hid at successive stages of development, and A08a dendrite length and branching were quantified (*Figure 6A*).

We controlled the onset of Hid expression in dbd using temperature-sensitive Gal80 (Gal80$^{ts}$). Hid expression was induced by shifting animals from 18°C to 30°C at 0 hr alh, 24 hr alh, or 48 hr alh and then assaying dendrite length at 72 hr alh (times adjusted to 25°C developmental equivalent). Our validation of Hid function in the embryo where Hid expression was initiated at stage 14 and ablation was observed by stage 17 (*Figure 3—figure supplement 1*) demonstrates that cell death can occur in this system within 9 hr. After lifting the Gal80 block on Hid expression at 48 hr alh, dbds were absent in an average of 65% of hemisegments 24 hr later (*Figure 6B–C*). These observations coupled with a previous report of Gal80$^{ts}$ inactivation kinetics (*McGuire et al., 2003*) suggest that dbd cell death occurs within 24 hr of shifting larvae to 30°C, and likely much sooner; however, we do not know the exact timing of dbd cell death with respect to each temperature shift.

A08a dendrite length was assayed for all experiments at 72 hr alh in A1 or A2 hemisegments that were innervated by 0–1 dbd neurons. If A08a retained the capacity for dendrite plasticity throughout larval life, we expected to detect increases in dendrite length and branching after ablating dbd at each timepoint. In contrast, compared to negative controls (*Figure 6D, F, H, J, L and M*), we found that A08a was competent to expand its lateral dendrites only after dbd was ablated in newly hatched larvae at 0 hr alh (*Figure 6F–G' and L–M*), but not when ablations occurred at 24 or 48 hr alh (*Figure 6H–M*). This result demonstrates that A08a dendrite structural plasticity is confined to an early critical period, perhaps the same critical period as used by the *Drosophila* larval motor system (*Ackerman et al., 2021*).

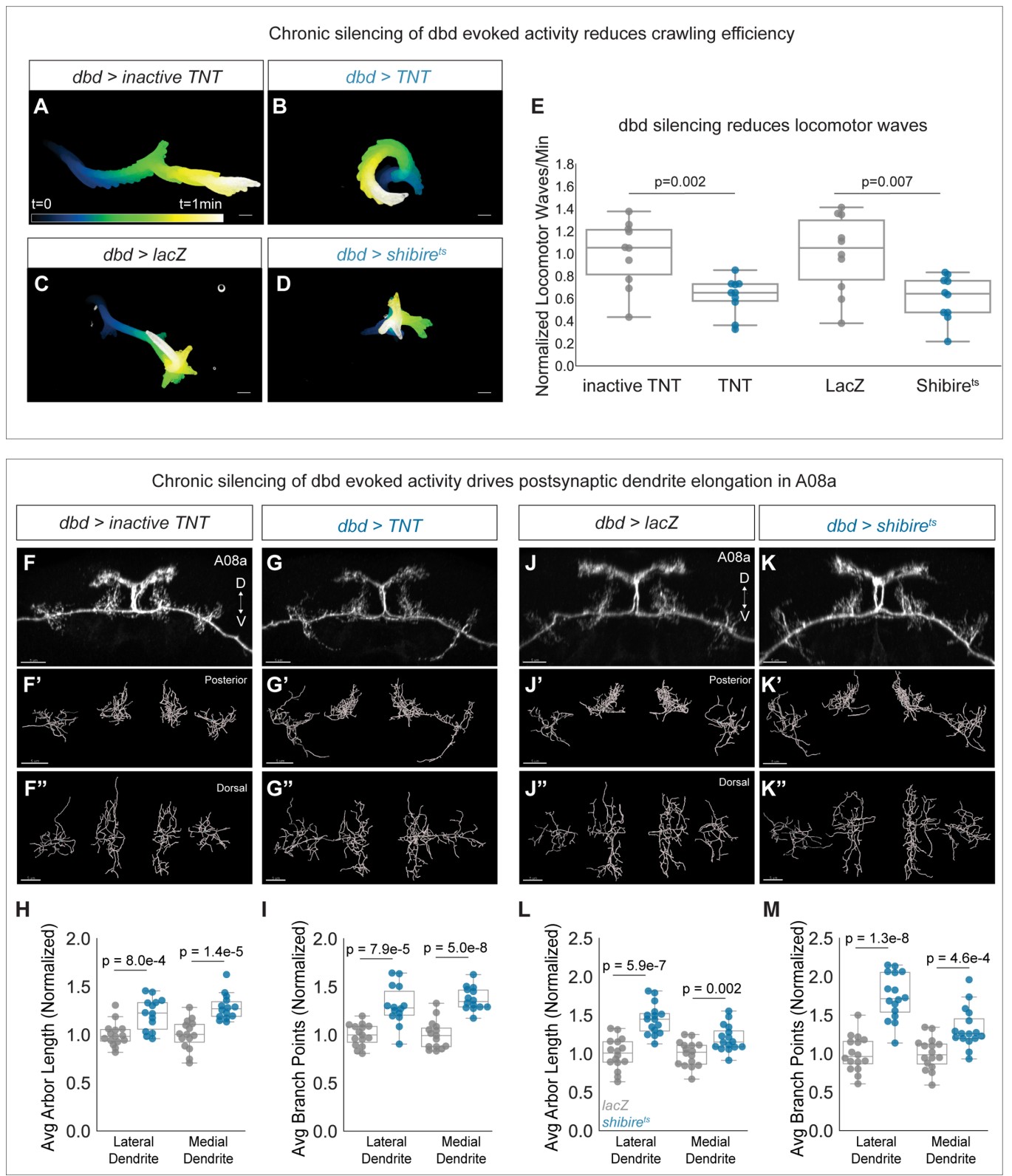

**Figure 4.** Chronic silencing of dbd activity drives A08a dendrite elongation. (**A**) Representative crawling trace of inactive tetanus toxin light chain (TNT) control larva (26±2 hr after larval hatching [alh]). Trace is color-coded by time. Scale bar, 1 mm. (**B**) Representative crawling trace of TNT larva (26±2 hr alh). (**C**) Representative crawling trace of control LacZ-expressing larva (24±2 hr alh). Trace is color-coded by time. Scale bar, 1 mm. (**D**) Representative crawling trace of Shibire[ts] larva (24±2 hr alh). (**E**) Number of locomotor waves (forward and reverse) initiated in 1 min, normalized to corresponding

*Figure 4 continued on next page*

*Figure 4 continued*

control. Control animals (gray) initiate more locomotor waves relative to animals with silenced dbd neurons (blue). Inactive TNT, n=10 animals; TNT, n=10 animals; LacZ, n=10 animals; Shibire[ts], n=10 animals (inactive TNT vs. TNT, p=0.002; LacZ vs. Shibire[ts], p=0.007). Circles represent locomotor waves of single animals. (**F**) Control A08a neurons at 26±2 hr alh, posterior view. (**F'**) Posterior view of Imaris filament reconstruction of A08a dendrites in (**F**). (**F"**) Dorsal view of (**F'**). (**G**) A08a neurons receiving input from TNT-expressing dbds at 26±2 hr alh, posterior view. (**G'–G"**) Imaris filament reconstructions of dendrites in (**G**). (**H**) Average dendrite length of lateral and medial A08a dendritic arbors in inactive TNT (control, gray, n=15 animals) or TNT-expressing animals (blue, n=14 animals) (Lateral Dendrite: p=8e-4; Medial Dendrite: p=1.4e-5). (**I**) Number of dendrite branch points of lateral and medial A08a dendritic arbors in inactive TNT (control, gray, n=15 animals) or TNT-expressing animals (blue, n=14 animals) (Lateral Dendrite: p=7.9e-5; Medial Dendrite: p=5e-8). (**J**) Control A08a neurons at 24±2 hr alh, posterior view. (**J'–J"**) Imaris filament reconstructions of dendrites in (**J**). (**K**) A08a neurons from receiving input from Shibire[ts]-expressing dbd, posterior view. (**K'–K"**) Imaris filament reconstructions of dendrites in (**K**). (**L**) Average dendrite length of lateral and medial A08a dendritic arbors in LacZ (control, gray, n=16 animals) or Shibire[ts]-expressing animals (blue, n=16 animals) (Lateral Dendrite: p=5.9e-7; Medial Dendrite: p=0.002). (**M**) Number of dendrite branch points of lateral and medial A08a dendritic arbors in LacZ (control, gray, n=16 animals) or Shibire[ts]-expressing animals (blue, n=16 animals) (Lateral Dendrite: 1.3e-8; Medial Dendrite: 4.6e-4). Images in F–G" and J–K" are Imaris 3D projections, Scale bars, 5 μm. Statistics computed using two-tailed unpaired t-test with unequal variance.

## Discussion
### Presynaptic contact promotes local dendrite outgrowth

We sought to uncover the cellular mechanism by which dendrites respond to variable positioning and input of their synaptic partners. Here and in our previous work, we rerouted the dbd axon terminal to the lateral and intermediate neuropils and found that A08a dendrites mirrored the location of their displaced presynaptic partner (*Valdes-Aleman et al., 2021*). When dbd was targeted to the A08a intermediate dendritic domain, an area devoid of dendrites in wild-type, ectopic branches were established. At the same time, we observed a decrease in medial dendrite volume and branching when dbd was targeted elsewhere. In these experiments, we measured no significant change to dbd-A08a functional connectivity strength (*Sales et al., 2019*), indicating that these compensations in dendrite length were likely activity-independent and due to contact alone.

Across a variety of model systems, presynaptic contact is correlated with or promotes the local outgrowth of dendrites (*Chen et al., 2010*; *Jacoby and Kimmel, 1982*; *Kamiyama et al., 2015*; *Niell et al., 2004*; *Vaughn, 1989*). In classic studies performed on the giant Mauthner (M) cells in zebrafish and axolotl, ablation of sensory afferents resulted in failed M-cell dendrite formation, whereas suprainnervation by these afferents was sufficient to cause overelaboration of the M-cell dendrites (*Goodman and Model, 1988*; *Kimmel et al., 1981*; *Kimmel et al., 1977*). Interestingly, when paired with a pharmacological silencing manipulation, suprainnervation of the M-cell still resulted in elongated dendrites, suggesting that contact-based cues are sufficient to drive local dendrite outgrowth (*Goodman and Model, 1990*); our results are consistent with these findings. Our similar results when dbd is mistargeted suggest a conserved mechanism across vertebrates and invertebrates for the local initiation of dendrites by sensory afferents. The ability for an axon to promote local dendrite outgrowth offers a potential strategy for pre- and postsynaptic partner matching that is robust to variable axon positioning.

One outstanding question from our studies is whether the dbd axon induces A08a dendrite formation de novo or selectively stabilizes nascent dendrites. There are currently no methods for tracking the initial formation of A08a dendrites. The LexA driver used to label A08a is first expressed in early larval life, after first outgrowth of A08a medial and lateral dendrites (*Figure 1D*). At the time of dbd CNS innervation at embryonic stage 14, the morphology of A08a dendrites is unknown. Therefore, it is unclear if dbd induces novel dendrite outgrowth from an otherwise bare neurite, or if it stabilizes and promotes continued dendrite elongation from an arbor already beginning to take form. In wild-type animals, A08a dendrites are innervated by multiple neurons (*Table 2*; *Sales et al., 2019*). If dbd is not the first neuron to contact the A08a medial dendritic domain, it is possible that the normal function of the dbd axon is to promote the continuous elongation of the medial dendrite rather than its initial induction. Either possibility would support our finding that presynaptic contact promotes postsynaptic dendrite outgrowth and clarifying the exact mechanism will be important for identifying the molecular players that support either process.

Until tools are developed that provide genetic access to A08a in early development, previous work on *Drosophila* dendrite development can give clues about the molecular mechanisms used to facilitate local dendrite outgrowth between dbd and A08a. For example, *Kamiyama et al., 2015*

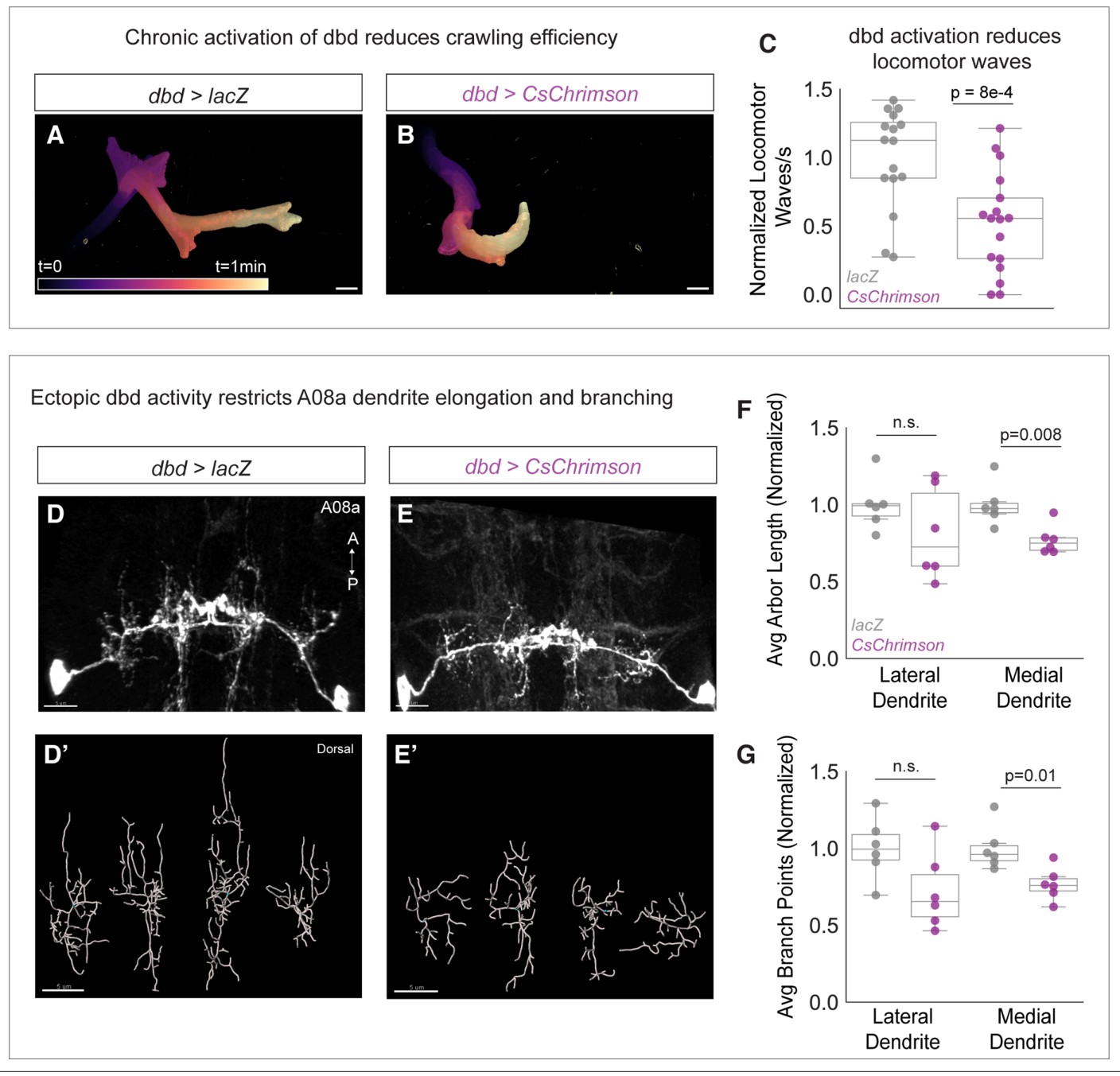

**Figure 5.** Chronic activation of dbd reduces A08a dendrite length. (**A**) Representative crawling trace of control (UAS-lacZ) larva (25±2 hr after larval hatching [alh]). Trace is color-coded by time. Scale bar, 1 mm. (**B**) Representative crawling trace of CsChrimson larva (25±2 hr alh). (**C**) Average number of locomotor waves (forward and reverse) per second, normalized to control average. Control animals (gray) initiate more locomotor waves relative to animals with Chrimson-activated dbd neurons (magenta) (p=8e-4). Control, n=16 animals; Chrimson, n=17 animals. (**D**) Control A08a neurons control animal at 25±2 hr alh, dorsal view. (**D'**) Imaris filament reconstructions of dendrites in (**D**). (**E**) A08a neurons receiving input from CsChrimson-expressing dbd neurons, activated throughout development. (**E'**) Imaris filament reconstructions of dendrites in (**E**). (**F**) Average dendrite length of lateral and medial A08a dendritic arbors in control (gray, n=6 animals) or CsChrimson-expressing animals (magenta, n=6 animals) (Lateral Dendrites: p=0.22; Medial Dendrites: p=0.008). (**G**) Number of dendrite branch points of lateral and medial A08a dendritic arbors in control (gray, n=6 animals) or CsChrimson-expressing animals (magenta, n=6 animals) (Lateral Dendrites: p=0.06; Medial Dendrites: p=0.01). Images are Imaris 3D projections. Scale bars, 5 μm. Values for all quantification normalized to control mean for either lateral or medial arbor. Circles represent single-animal averages between left and right hemisegments. Statistics computed using two-tailed unpaired t-test with unequal variance. n.s.=not significant, p>0.05.

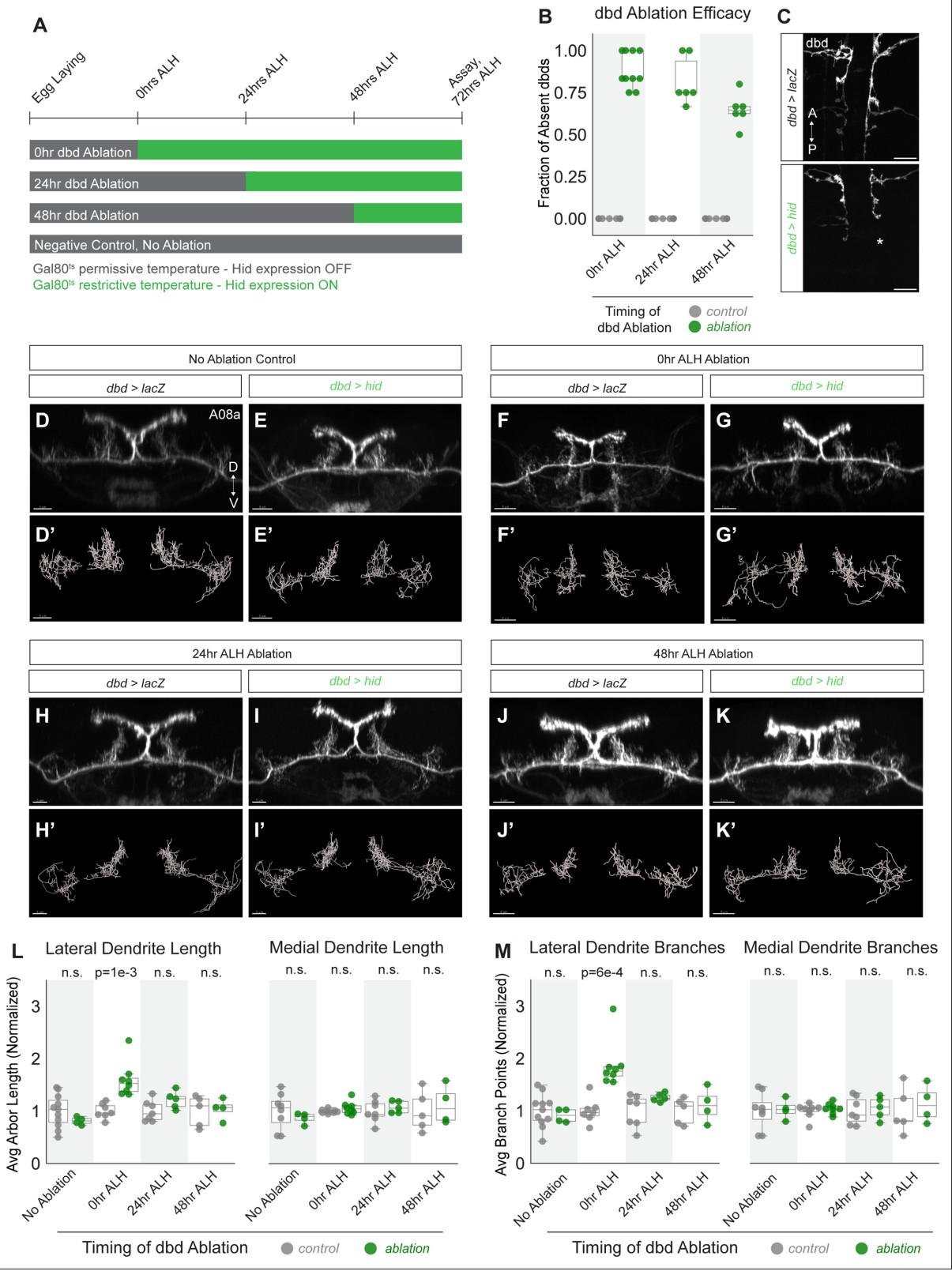

**Figure 6.** A08a dendrite plasticity is confined to a critical period in larval development. (**A**) Experimental design. Hid expression was inhibited at 18°C by Gal80ts (gray bars). Hid expression was induced by shifting animals to 30°C (green bars) at the 25°C equivalent of 0 hr after larval hatching (alh), 24 hr alh, and 48 hr alh. A08a dendrites were assayed for length and branching at the 25°C developmental equivalent of 72 hr alh. (**B**) Fraction of dbds absent in control (UAS-lacZ) or ablation (UAS-hid) samples at 72 hr alh after initiating Hid expression at 0 hr alh (control n=11 animals; hid n=10 animals),

*Figure 6 continued on next page*

*Figure 6 continued*

24 hr alh (control n=15 animals, Hid n=6 animals), and 48 hr alh (control n=13 animals, Hid n=6 animals). (**C**) Representative images of control (top) and ablation (bottom) samples showing first two abdominal segments. Samples are from the 48 hr alh ablation cohort, and were imaged at 72 hr alh. Control animals retain the full complement of dbds, while the hid sample shows a missing dbd in A2 indicated by the asterisk (*). Scale bars, 10 µm. (**D**) Control A08a raised continuously at 18°C. Imaris filament reconstructions of A08a dendrites in (**D'**). (**E**) A08a in hid-expressing animal raised continuously at 18°C. Imaris filament reconstructions of A08a dendrites in (**E'**). (**F**) A08a in control animal shifted to 30°C at 0 hr alh. Imaris filament reconstructions of A08a dendrites in (**F'**). (**G**) A08a in hid-expressing animal shifted to 30°C at 0 hr alh. Imaris filament reconstructions of A08a dendrites in (**G'**). (**H**) A08a in control animal shifted to 30°C at 24 hr alh. Imaris filament reconstructions of A08a dendrites in (**H'**). (**I**) A08a in hid-expressing animal shifted to 30°C at 24 hr alh. Imaris filament reconstructions of A08a dendrites in (**I'**). (**J**) A08a in control animal shifted to 30°C at 48 hr alh. Imaris filament reconstructions of A08a dendrites in (**J'**). (**K**) A08a in hid-expressing animal shifted to 30°C at 48 hr alh. Imaris filament reconstructions of A08a dendrites in (**K'**).Images in D-K are Imaris 3D projections. Scale bars, 5µm. (**L**) Average dendrite length of lateral (left) and medial (right) A08a dendritic arbors in control (UAS-lacZ; gray) or experimental, ablation (UAS-hid, green). (**M**) Average number of branch points on lateral (left) and medial (right) A08a dendrites in control (UAS-lacZ; gray) or experimental, ablation (UAS-hid, green). X-axes, timing of dbd ablation (no ablation control: control n=8–11 animals, Hid n=4 animals; 0 hr alh ablation: control n=7–8 animals, Hid n=8 animals; 24 hr alh ablation: control n=7 animals, Hid n=5 animals; 48 hr alh ablation: control n=5 animals, Hid n=4 animals). For Hid quantifications, only segments containing 0–1 dbds were analyzed (determined by absence of dbd membrane stain). Values for all quantification normalized to control mean for each ablation timepoint. Circles represent single-animal averages between left and right hemisegments. Statistics computed using two-tailed unpaired t-test with unequal variance. n.s=not significant, p>0.05.

previously found that the initiation of aCC dendrite outgrowth coincides with when the cell receives presynaptic input from the pioneering interneuron MP1. MP1 expresses the Dscam1 cell adhesion molecule; presynaptic Dscam1 promotes the postsynaptic accumulation of Pak1 and Cdc42 in aCC, specifying the precise location of aCC dendrite elongation (*Kamiyama et al., 2015*; *Wilhelm et al., 2022*). Perhaps a similar transsynaptic cell adhesion mechanism underlies ectopic A08a dendrite formation when dbd is targeted to the intermediate zone between lateral and medial dendrites.

## Presynaptic activity inhibits dendrite outgrowth neuron-wide

Homeostatic structural plasticity is a phenomenon in which neurites adjust their length to counter the effect of too much or too little activity; when activity is excessive, the dendrite shrinks, and when activity is diminished, the dendrite expands. Homeostatic regulation of dendritic arbor size has been documented in insects (*Ackerman et al., 2021*; *Hoy et al., 1985*; *Tripodi et al., 2008*; *Yuan et al., 2011*) and vertebrates (*Shen et al., 2020*; *Takeo et al., 2021*; *Tanvir et al., 2021*; *Wu and Cline, 1998*). We observed a homeostatic relationship between presynaptic activity levels and postsynaptic A08a dendrite length. When synaptic input onto A08a was decreased by silencing evoked dbd activity, A08a dendrites were elongated; when dbd was chronically activated, A08a dendrites were smaller. In contrast to the robust effect of inhibiting dbd neurotransmission, A08a dendrite length was modestly diminished in response to dbd stimulation. One reason for this could be that the inhibitory A02d neuron, which is a top input to A08a (*Table 2*), also receives input from dbd (*Table 1*). A02d could therefore potentially cancel out some effects of dbd excitation in A08a. Additionally, while CsChrimson has previously been used to chronically activate neurons for minutes to hours (*Ackerman et al., 2021*), it is not clear that the channel actually remains open for such long periods of time. Compensatory adjustments in dendrite length are likely a strategy to maintain a 'set-point' of synaptic input. Such a mechanism would be useful to maintain a constant level of postsynaptic output when the amount of input is variable.

When dbd sensory innervation to the A08a medial dendrite was lost due to dbd ablation, A08a lateral dendrites were elongated whereas the medial arbor was maintained. We hypothesize that when dbd is ablated, A08a medial dendrites are unchanged due to the opposing roles of presynaptic activity and presynaptic contact. Lack of dbd contact (which restricts growth) and lack of dbd activity (which promotes growth) could be working in direct opposition and resulting in no change in medial arbor size. In contrast, the lateral arbor experiences only the loss of activity (which acts neuron-wide), leading to arbor growth. Taken together, our data support a model of dendrite development in which presynaptic contact acts locally to promote dendrite outgrowth, whereas presynaptic activity acts globally across an entire neuron to downregulate dendrite elongation (*Figure 7*). We propose that as synapses are added and become functional, activity could act as a negative-feedback mechanism to curb excessive dendrite elongation across the postsynaptic neuron.

How might activity limit dendrite growth throughout the postsynaptic neuron? Sustained alterations to neurotransmission would likely impact postsynaptic calcium levels and therefore intracellular

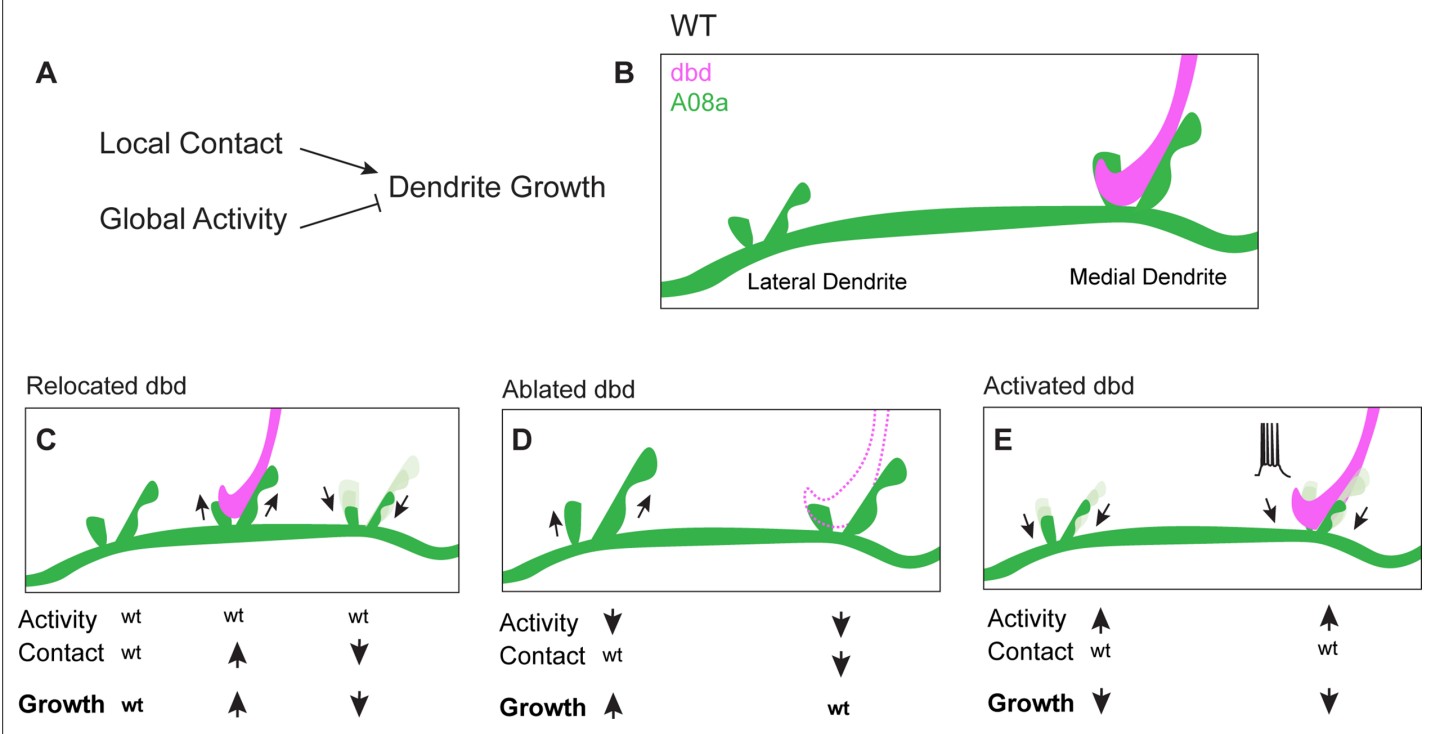

**Figure 7.** Proposed model: Presynaptic activity and contact opposingly regulate dendrite outgrowth. (**A**) Presynaptic contact promotes local dendrite outgrowth, while presynaptic activity levels inhibit neuron-wide dendrite outgrowth. (**B**) Wild-type (WT) A08a dendrites. (**C**) When dbd is mistargeted to intermediate A08a dendritic domain, activity levels are wt. Local outgrowth is promoted at the intermediate domain, and inhibited by lack of contact at the medial dendrite. (**D**) When dbd is ablated, neuron-wide activity levels are decreased. This promotes lateral dendrite outgrowth. Lack of contact at the medial dendrite opposes the activity-dependent drive to elongate so dendrite length remains wt. (**E**) dbd activation promotes neuron-wide dendrite retraction/premature stabilization.

calcium signaling. Indeed, studies in *Drosophila* show PKA regulation of dendritic arbor size (*Baines, 2003*; *Tripodi et al., 2008*). Calcium can also signal through regulators of transcription, allowing the effect of activity to potentially be exerted neuron-wide through changes in gene expression (*Sun et al., 1994*). While calcium signaling pathways have probably been the most explored among activity-dependent mechanisms regulating dendrite growth, it is likely one of many pathways that the cell could deploy. For example, Wnt signaling is also recruited to inhibit dendrite elongation when activity levels are high (*Singh et al., 2010*). In the future, it will be interesting to explore the context in which each type of signaling pathway is engaged, and how they could oppose the effect of growth-promoting contact.

## A critical period for dendrite development in the *Drosophila* larva

Critical periods are developmental windows of heightened neural plasticity that are present from vertebrates (*Kalb, 1994*; *Keck et al., 2017*; *Takesian and Hensch, 2013*; *Walton et al., 1992*) to insects (*Caglayan and Gilbert, 1987*; *Elekonich et al., 2003*; *Hartfelder et al., 1995*; *Levine et al., 1986*; *Morgan et al., 1998*). In *Drosophila*, critical periods have been identified in the late embryo/ early larva (*Ackerman et al., 2021*; *Crisp et al., 2011*; *Fushiki et al., 2013*; *Giachello et al., 2021*; *Giachello and Baines, 2015*; *Hartwig et al., 2008*; *Jarecki and Keshishian, 1995*; *Tripodi et al., 2008*) and in the newly eclosed adult (*Doll et al., 2017*; *Doll and Broadie, 2016*; *Doll and Broadie, 2015*; *Golovin et al., 2021*; *Golovin et al., 2019*). Altered activity during the critical period can lead to long-lasting defects in neuronal morphology and behavior (*Ackerman et al., 2021*; *Crisp et al., 2011*; *Fushiki et al., 2013*; *Giachello et al., 2021*; *Giachello and Baines, 2015*; *Hartwig et al., 2008*; *Jarecki and Keshishian, 1995*; *Tripodi et al., 2008*). In the *Drosophila* embryonic motor system, dendrite length can be homeostatically modified by levels of activity (*Tripodi et al., 2008*). The motor dendrites lose the capacity to undergo activity-dependent remodeling at 8 hr alh, early in larval life. The closure of this critical period is governed by astrocytes – their infiltration into the

neuropil coincides with critical period closure and their contact with motor dendrites prevents precocious dendrite extension/retraction (*Ackerman et al., 2021*; *Stork et al., 2014*).

Despite the clear evidence for this early window of plasticity in the larva, it was unclear whether activity-dependent structural plasticity could be re-engaged in later stages of larval life. The larval nervous system continues to grow throughout its life, with individual neurons adding hundreds of microns of overall dendrite length (*Gerhard et al., 2017*). We therefore wondered if an interneuron such as A08a would be subject to the same critical period for structural plasticity described for larval motor neuron dendrites, or if dendrite plasticity remains necessary in subsequent stages of larval development to accommodate animal growth. Our findings showed that only ablation of dbd prior to 24 hr alh (and thus loss of its excitatory input to A08a) increases A08a dendrite length. This timing is similar to the critical period for motor dendrite structural plasticity (*Ackerman et al., 2021*). These results suggest that larval VNC neurons may generally be subject to the same early critical period regulated by astrocytes, and structural plasticity events may not be inducible in later stages of development. If presynaptic input is not required to regulate dendrite outgrowth in subsequent stages of larval life, perhaps separate cell-intrinsic (*Tenedini et al., 2019*; *Zwart et al., 2013*) or mechanical mechanisms (e.g. stretch or pulling forces) (*Balice-Gordon and Lichtman, 1990*; *Bray, 1984*; *Tao et al., 2022*) are required to allow the scaling of circuits as an organism grows.

### Dendrite diversification as a substrate for behavioral evolution

Here and in previous work, we observed that A08a dendrite development is not hard-wired, but is modulated by the location of presynaptic partners and presynaptic activity levels (*Sales et al., 2019*; *Valdes-Aleman et al., 2021*). Within a species, subtle variation in neuron morphology that arises in development can diversify behavior. For example, natural variation in *Drosophila melanogaster* dorsal cluster neuron axonal projections directly impacts an animal's ability to orient toward a visual object (*Linneweber et al., 2020*). Imprecise but constrained morphological development could be a 'bet-hedging' strategy to promote the adaptability of an individual species to environmental changes.

Across species, it is intriguing to speculate that species-specific behaviors evolved in part through mechanisms that pattern neurite morphology. For two highly divergent nematode species, *Caenorhabditis elegans* and *Pristionchus pacificus*, amphid sensilla neuron number and soma position are highly conserved whereas ciliated dendrite morphology is more diverse. Correspondingly, neurons whose structure is more dissimilar between the species also have more divergent synaptic connections, implying that downstream behaviors would also differ (*Hong et al., 2019*). Another study found that in a species of *Drosophila* with evolved attraction to noni fruit, axon branch morphology in an olfactory processing center diverged from that of species that do not exhibit attraction to noni (*Auer et al., 2020*). For interneurons such as A08a, genomic changes resulting in alterations to dendrite neuropil position could vary the number and identities of presynaptic partners. It will be interesting to causally test the impact of neurite architecture on behavioral diversification, and specifically the extent to which genes regulating presynaptic axon position, neural activity, or critical period length are nodes of dendritic and ultimately behavioral evolution.

## Materials and methods

**Key resources table**

| Reagent type (species) or resource | Designation | Source or reference | Identifiers | Additional information |
|---|---|---|---|---|
| Strain, strain background (*Drosophila melanogaster*) | 26F05-LexA | BDSC | 54702 | Expressed in A08a neurons |
| Strain, strain background (*Drosophila melanogaster*) | 165-Gal4 | W Grueber | N/A | Expressed in dbd neurons, as well as clVda neurons |
| Strain, strain background (*Drosophila melanogaster*) | UAS-robo2::HA | BDSC | 66886 | Expresses HA-tagged Robo2 under UAS control |
| Strain, strain background (*Drosophila melanogaster*) | UAS-unc-5::HA | A Zarin | N/A | Expresses HA-tagged Unc-5 under UAS control |

*Continued on next page*

*Continued*

| Reagent type (species) or resource | Designation | Source or reference | Identifiers | Additional information |
|---|---|---|---|---|
| Strain, strain background (*Drosophila melanogaster*) | *UAS-hid, tubP-Gal80ts/CyO* | M Freeman | N/A | Expresses Hid under UAS control, Gal80ts negatively regulates Gal4 at 18°C. |
| Strain, strain background (*Drosophila melanogaster*) | *UAS-TeTxLC-G2* | BDSC | 28838 | Expresses the light chain of tetanus toxin under UAS control |
| Strain, strain background (*Drosophila melanogaster*) | *UAS-inactive TeTxLC* | BDSC | 28840 | Expresses a mutated tetanus toxin light chain gene under UAS control |
| Strain, strain background (*Drosophila melanogaster*) | *UAS-shi[ts1].K* | BDSC | 44222 | Expresses temperature-sensitive shi under UAS control for inhibiting synaptic transmission at 30°C |
| Strain, strain background (*Drosophila melanogaster*) | *UAS-LacZ* | BDSC | 8529 | Control transgene |
| Strain, strain background (*Drosophila melanogaster*) | *UAS-LacZ* | BDSC | 8530 | Control transgene |
| Strain, strain background (*Drosophila melanogaster*) | *10xUAS-IVS-myr::smGdP::HA, 13xLexAop2-IVS-myr::smGdP::V5* | BDSC | 64092 | Expresses HA membrane tag under UAS control, V5 membrane tag under LexAop control |
| Strain, strain background (*Drosophila melanogaster*) | *UAS-IVS-CsChrimson.mVenus (attP2)* | V Jayaraman | N/A | Expresses CsChrimson tagged with mVenus under UAS control |
| Antibody | Anti-V5 tag mouse monoclonal | Invitrogen, Carlsbad, CA | Cat. R96025, Lot 1949337 | (1:1000) |
| Antibody | Anti-HA tag rat monoclonal | Roche Holding, AG, Basel, Switzerland | Cat. 11867423001, Lot 27573500 | (1:100, after suggested dilution) |
| Antibody | Anti-Futsch (22C10) mouse monoclonal | DSHB, Iowa City, IA | Concentrate 0.1 mL | (1:1000) |
| Antibody | Anti-Corazonin rabbit polyclonal | J Veenstra, Univ Bordeaux | N/A | (1:2000) |
| Antibody | Alexa Fluor 488 AffiniPure Donkey Anti-Mouse IgG (H+L) donkey polyclonal | Jackson ImmunoResearch, West Grove, PA | Cat. 715-545-151 | (1:400) |
| Antibody | Alexa Fluor 647 AffiniPure Donkey Anti-Rat IgG (H+L) donkey polyclonal | Jackson ImmunoResearch, West Grove, PA | Cat. 712-605-153 | (1:400) |
| Antibody | Alexa Fluor 488 AffiniPure Donkey donkey polyclonal Anti-Rat IgG (H+L) | Jackson ImmunoResearch, West Grove, PA | Cat. 712-545-153 | (1:400) |
| Antibody | Alexa Fluor RRX AffiniPure Donkey Anti-Mouse IgG (H+L) donkey polyclonal | Jackson ImmunoResearch, West Grove, PA | Cat. 715-295-151 | (1:400) |
| Antibody | Alexa Fluor 647 AffiniPure Donkey Anti-Mouse IgG (H+L) donkey polyclonal | Jackson ImmunoResearch, West Grove, PA | Cat. 715-605-151 | (1:400) |

## Fly stocks

| Genotypes | Figure |
|---|---|
| Females containing *10xUAS-IVS-myr::smGdP::HA, 13xLexAop2-IVS-myr::smGdP::V5* (BDSC# 64092); *GMR26F05-LexA* (A08a neurons) (BDSC# 54702), *UAS-bruchpilot (short)-mstraw*; *165*-Gal4 (dbd neurons) were in-crossed to males of the same genotype. | *Figure 1A–C* |

*Continued on next page*

*Continued*

| Genotypes | Figure |
|---|---|
| Females containing *10xUAS-IVS-myr::smGdP::HA, 13xLexAop2-IVS-myr::smGdP::V5* (BDSC# 64092); *GMR26F05-LexA* (A08a neurons) (BDSC# 54702), *UAS-bruchpilot (short)-mstraw*; *165*-Gal4 (dbd neurons) were crossed to males containing either (1) *UAS-lacZ* (BDSC #8529), (2) *UAS-robo2::HA* (BDSC #66886), or (3) *UAS-unc-5::HA* | 1. *Figure 2A and D–F* <br> 2. *Figure 2B and D–F* <br> 3. *Figure 2C–D* |
| Females containing *10xUAS-IVS-myr::smGdP::HA, 13xLexAop2-IVS-myr::smGdP::V5* (BDSC# 64092); *GMR26F05-LexA* (A08a neurons) (BDSC# 54702), *UAS-bruchpilot (short)-mstraw*; *165*-Gal4 (dbd neurons) were crossed to males containing either (1) *UAS-lacZ.Exel* (control) (BDSC# 8529) or (2) *UAS-hid, tubP-Gal80ts/CyO*. | 1. *Figure 3A–B' and E–G; Figure 6B, D, F, H, J and L–M; Figure 3—figure supplement 1B-C'.* <br> 2. *Figure 3C–G; Figure 6B-C, E, G, I, K-M; Figure 3—figure supplement 1B, D-D'.* |
| Females containing *10xUAS-IVS-myr::smGdP::HA, 13xLexAop2-IVS-myr::smGdP::V5* (BDSC# 64092); *GMR26F05-LexA* (A08a neurons) (BDSC# 54702), *UAS-bruchpilot(short)-mstraw*; *165*-Gal4 (dbd neurons) were crossed to males containing either (1) *UAS-inactive TeTxLC* (BDSC #28840) or (2) *UAS-TeTxLC-G2* (BDSC #28838) | 1. *Figure 4A, E-F", H-I* <br> 2. *Figure 4B, E, G-I* |
| Females containing *10xUAS-IVS-myr::smGdP::HA, 13xLexAop2-IVS-myr::smGdP::V5* (BDSC# 64092); *GMR26F05-LexA* (A08a neurons) (BDSC# 54702), *UAS-bruchpilot(short)-mstraw*; *165*-Gal4 (dbd neurons) were crossed to males containing either (1) *UAS-lacZ* (BDSC #8529) or (2) *UAS-shi[ts1].K* (BDSC #44222) | 1. *Figure 4C, E, J–J" and L–M* <br> 2. *Figure 4D–E and K–M* |
| Females containing *10xUAS-IVS-myr::smGdP::HA, 13xLexAop2-IVS-myr::smGdP::V5* (BDSC# 64092); *GMR26F05-LexA* (A08a neurons) (BDSC# 54702), *UAS-bruchpilot(short)-mstraw*; *165*-Gal4 (dbd neurons) were crossed to males containing (1) *UAS-lacZ* (BDSC #8530) or (2) *UAS-IVS-CsChrimson.mVenus* (attP2). | 1. *Figure 1D–E Figure 5A, C–D' and F–G* <br> 2. *Figure 5B–C and E–G* |

## Animal preparation

### Embryo experiments

#### Dbd-Gal4 expression (Figure 1)

Embryos were collected overnight for 16 hr on 3.0% agar apple juice caps with yeast paste at 25°C. Embryonic stages were identified post hoc by analyzing gut morphology. Stage 14, gut is tube shaped. Stage 15, gut is heart shaped. Stage 16, gut is coiled three times. Stage 17, gut is coiled four times.

#### Hid validation (Figure 3—figure supplement 1)

Embryos were collected 4 hr on 3.0% agar apple juice caps with yeast paste at 25°C and were then aged at 30°C for 11 hr until approximately stage 17.

#### A08a-LexA expression (*Figure 1*)

Embryos were collected on 3.0% agar apple juice caps with yeast paste for 4 hr at 25°C. Embryos were then aged for 21 hr. After 21 hr, embryos were transferred to a fresh 3.0% agar apple juice cap and then aged for 4 hr. Half of the hatched larvae were immediately dissected (aged 2±2 hr alh). The other half of the larvae were transferred to standard cornmeal fly food dishes and aged an additional 24 hr until dissection at 26±2 hr alh. Due to stochastic expression of A08a-LexA in newly hatched larvae, samples were stained for Corazonin as a VNC segment landmark. Corazonin lables cells in T2-A6 (*Choi et al., 2005*).

#### TNT experiments (*Figure 4*)

Embryos were collected on 3.0% agar apple juice caps with yeast paste for 4 hr at 25°C. Embryos were then aged for 21 hr. After 21 hr, embryos were transferred to a fresh 3.0% agar apple juice cap and

then aged for 4 hr. Hatched larvae were transferred to standard cornmeal fly food dishes and aged until dissection at 26±2 hr alh.

### Shibire[ts] and Hid experiments (*Figure 3*, *Figure 3—figure supplement 1* and *Figure 4*)

Embryos were collected on 3.0% agar apple juice caps with yeast paste for 4 hr at 25°C. Embryos were then aged for 17 hr at 30°C (embryos and larvae develop 1.23× faster at 30°C). After 17 hr, embryos were transferred to a fresh 3.0% agar apple juice cap and then aged for 4 hr. Hatched larvae were transferred to standard cornmeal fly food dishes and aged at 30°C until dissection at 24±2 hr alh.

### Chrimson experiment (*Figure 5*)

All-*trans* retinal (ATR) is a necessary co-factor for CsChrimson. To ensure maternal transfer of ATR to larval progeny, parental crosses were fed yeast paste supplemented with ATR (final concentration 0.5 mM; Sigma-Aldrich, R2500-100MG) for 72 hr. ATR yeast was made fresh daily and kept away from light. Embryos were then collected on 3.0% agar apple juice caps with +ATR yeast paste for 4 hr at 25°C. Embryos and larvae were aged continuously under broad spectrum light, approximately 10 cm from the light source (~30,000 lux; measured using Light Meter app for iPhone – Lightray Innovation GmbH). The temperature under the light was 28°C. Embryos age 1.1× faster at 28°C. After 19 hr, embryos were transferred to a fresh 3.0% agar apple juice cap and then aged for 4 hr. Newly hatched larvae were collected after the 4 hr and reared for an additional 23 hr under the light (larvae age 1.03× faster 28°C). Larvae were dissected at 25±2 hr in low light (<100 lux) to prevent further Chrimson activation.

### Critical period experiments (*Figure 6*)

Embryos were collected on 3.0% agar apple juice caps with yeast paste for 4 hr at 25°C.

#### No ablation group

Embryos were aged for 42 hr at 18°C (embryos and larvae develop 2× slower at 18°C). After 42 hr, embryos were transferred to a fresh 3.0% agar apple juice cap and then aged for 4 hr. Hatched larvae were transferred to standard cornmeal fly food dishes and aged at 18°C until dissection at 146±2 hr alh (25°C equivalent to 72 hr alh, middle of third instar).

#### 0 hr alh ablation group

Embryos were aged for 42 hr at 18°C. After 42 hr, embryos were transferred to a fresh 3.0% agar apple juice cap and then aged for 4 hr at 30°C. Hatched larvae were transferred to standard cornmeal fly food dishes and aged at 30°C until dissection at 67±2 hr alh (25°C equivalent to 72 hr alh, middle of third instar).

#### 24 hr alh ablation group

Embryos were aged for 42 hr at 18°C. After 42 hr, embryos were transferred to a fresh 3.0% agar apple juice cap and then aged for 4 hr. Hatched larvae were transferred to standard cornmeal fly food dishes and aged at 18°C for 48 hr. At this time animals were shifted to 30°C and raised an additional 44 hr until the time of dissection (25°C equivalent to 72 hr alh, middle of third instar).

#### 48 hr alh ablation group

Embryos were aged for 42 hr at 18°C. After 42 hr, embryos were transferred to a fresh 3.0% agar apple juice cap and then aged for 4 hr. Hatched larvae were transferred to standard cornmeal fly food dishes and aged at 18°C for 96 hr. At this time animals were shifted to 30°C and raised an additional 22 hr until the time of dissection (25°C equivalent to 72 hr alh, middle of third instar).

No groups could be tested in which animals were reared continuously at 30°C until third instar as all animals expressing dbd > Hid died.

## Immunohistochemistry

### Larval brain sample preparation

Larval brains were dissected in PBS, mounted on pre-EtOH treated 12 mm #1thickness poly-D-lysine-coated coverslips (Neuvitro Corporation, Vancouver, WA, Cat# GG-12-PDL) (primed in 70% EtOH at least 1 day prior to use). Samples fixed for 23 min in fresh 4% paraformaldehyde (PFA) (Electron Microscopy Sciences, Hatfield, PA, Cat. 15710) in PBST. Samples were washed in 0.3% PBST and then blocked with 2% normal donkey serum and 2% normal goat serum (Jackson ImmunoResearch Laboratories, Inc, West Grove, PA) in PBST overnight at 4°C or for 1 hr at room temperature. Samples incubated in primary antibody for 2 days at 4°C. The primary was removed, and the samples were washed with two quick PBST rinses followed by 3×20 min washes in PBST. Samples were then incubated in secondary antibodies overnight at 4°C, shielded from light. The secondary antibody was removed following overnight incubation and the brains were washed in PBST (two quick rinses, followed by 3×20 min washes). Samples were dehydrated with an ethanol series (30%, 50%, 75%, 100% ethanol; all v/v, 10 min each) (Decon Labs, Inc, King of Prussia, PA, Cat. 2716GEA) then incubated in xylene (Fisher Chemical, Eugene, OR, Cat. X5-1) for 2×10 min. Samples were mounted onto slides containing 2 drops of DPX mountant (MilliporeSigma, Burlington, MA, Cat. 06552) and cured for 1–3 days then stored at 4°C until imaged.

### Embryo sample preparation

Embryos were transferred from apple caps into collection baskets and rinsed with dH$_2$O. Embryos were dechorionated in 100% bleach (Clorox, Oakland, CA) for 3 min and 30 s with gentle agitation. Dechorionated embryos were rinsed with dH$_2$O for 1 min. Embryos were fixed 25 min in 2 mL Eppendorf tubes containing equal volumes of heptane (Fisher Chemical, Eugene, OR, H3505K-4) and 4% PFA diluted in PBS. Fix was removed, and 850 µL of heptane was added to each tube; 650 µL of methanol were then added, and tubes were then subject to vigorous agitation for 1 min in a step required for removing the vitelline membrane. Nearly all liquid was removed from the tubes, leaving the embryos. Embryos were rinsed in methanol (Fisher Chemical, Eugene, OR, Lot# 206197, Cat. A412P-4) twice followed by two quick rinses in 0.3% PBST. PBST was removed and embryos were blocked with 2% normal donkey serum and 2% normal goat serum (Jackson ImmunoResearch Laboratories, Inc, West Grove, PA) in PBST for 1 hr at room temperature. After blocking, embryos were incubated in primary and secondary antibodies and mounted as described above for larval brains.

## Light microscopy

Fixed larval preparations were imaged with a Zeiss LSM 710 or LSM 900 laser scanning confocal (Carl Zeiss AG, Oberkochen, Germany) equipped with an Axio Imager.Z2 microscope. A 63×/1.40 NA Oil Plan-Apochromat DIC m27 objective lens and GaAsP photomultiplier tubes were used. Software program used was Zen 2.3 (blue edition) (Carl Zeiss AG, Oberkochen, Germany). For each independent experiment, all samples were acquired using identical acquisition parameters.

## Image processing and analysis

### Imaris filament reconstruction and quantification of A08a dendrites (*Figures 3–6*)

Confocal image stacks were loaded into Imaris 9.5.1 (Bitplane AG, Zurich, Switzerland). A08a dendrites from A1 and A2 segments were analyzed. A new Imaris filament object was created for each A08a dendrite (lateral and medial). Briefly, the Filaments tool was selected, and a region of interest (ROI) drawn to encompass the dendrite. The source channel for A08a membrane (488) was selected. An approximation for minimum and maximum dendrite diameters were measured in Slice view, and found to be 0.2 and 1 µm, respectively. These values were used to identify Starting Points and Seed Points for all images. Thresholds for Starting Points and Seed Points were manually adjusted until 1 Starting Point on the main A08a dendritic shaft remained, and Seed Points labeled A08a dendrite signal without labeling image background. The option to Remove Disconnected Seed Points was

selected, with a Smoothing factor of 0.2 μm. Absolute Intensity Threshold was manually adjusted until all Seed Points were filled. When the Filament was rendered, misidentified structures were selected and manually deleted or adjoined.

The sum length and the sum of all branch points of each dendritic arbor were calculated automatically in Imaris (Statistics > Details > Average Values). The values for left/right lateral dendrites and left/right medial dendrites were averaged for each animal.

For *Figure 3G*, the length of the A08a Left-Lateral Dendrite, Left-Medial Dendrite, Right-Medial Dendrite, Right-Lateral Dendrite was summed together, as left and right A08a's form recurrent synaptic connections. The values were normalized to the mean total dendrite length of the wild-type controls (with 4 dbd inputs).

Sample exclusion criteria were established prior to conducting analysis. Samples were excluded from analysis if there was damage to the tissue or low signal to noise that obstructed the ability to reliably identify dendrite membrane signal. For critical period experiments in which dbd was ablated using Hid (*Figure 6*), only segments innervated by 0–1 dbds (i.e. all dbds innervating the segment were ablated, or all but 1 dbd was ablated) were used for quantifying A08a dendrite morphology.

## A08a cumulative dendrite position (*Figure 2D*)

Published data from *Sales et al., 2019*, and *Valdes-Aleman et al., 2021*, were used. Larval brains aged 24±4 hr alh were processed and imaged as described in *Sales et al., 2019*, and *Valdes-Aleman et al., 2021*. Briefly, image processing and analysis was performed using FIJI (ImageJ 1.50d, https://imagej. net/Fiji). Stepwise, images were rotated (Image > Transform > Rotate(bicubic)) to align dendrites of interest along the x-axis, then a standardized ROI was selected in 3D to include the dendrites to analyze in one hemisegment (Rectangular selection > Image > Crop). To identify the voxels that contain dendrite intensity, a mask was manually applied (Image > Adjust > Threshold). The threshold was assigned to include dendrite positive voxels and minimize contribution from background. To quantify the amount of dendrite positive voxels across the medial-lateral axis, images were reduced in the z-dimension (Image > Stacks > Z-project > Sum Slices) and a plot profile was obtained to measure the average voxel intensity (Rectangular selection > Analyze > Plot profile). The cumulative sum of A08a dendrite voxels was calculated for each individual hemisegment, and a mean voxel distribution was generated from the population data.

## Branch distribution (*Figure 2E-F*)

Published data from *Sales et al., 2019*, and *Valdes-Aleman et al., 2021*, were used. Hemisegments from A1 and A2 were analyzed. Image analysis was performed in FIJI. Images were rotated to align the dendrites of interest along the x-axis (Image > Transform > Rotate(bicubic)). For each hemisegment analyzed, a rectangular ROI was drawn starting at the midline and ending at the lateral edge of the lateral-most branch point coming from the main A08a neurite. The width of the ROI was logged (in microns). The Cell Counter plugin was used to count the dendrite branches originating from the main A08a neurite (Plugins > Analyze > Cell Counter). The x position of each branch point was measured (in microns) (Cell Counter > Measure). The relative lateral-medial position of each branch point was determined by dividing the branch's x position by ROI width. The relative frequency of branches at a given position was determined by counting the number of branches for that bin, and dividing that value by the total number of branches.

For *Figure 2F*, Lateral, Intermediate, and Medial domains were determined based on the relative peak branch positions in LacZ controls in *Figure 2E*. The Lateral domain = <0.35, Intermediate = ≥0.35 and<0.5, and Medial = >0.5. The proportion of branches in each domain was determined for each cell and was then plotted as a single point in *Figure 2F*.

## Validation of Hid function and criteria for sample inclusion

For *Figure 3—figure supplement 1*, only hemisegments with clear 22C10 labeling were analyzed for the presence or absence of dbd. In the 22C10 pattern, the dbd cell body can be identified as the basal-most cell body among the dorsal cluster of neurons in the body wall, along with its characteristic bipolar dendrites. For an individual animal, the total number of dbds was manually counted and divided by the total number of 22C10-labled hemisegments to obtain a percentage of dbds present in controls or ablation animals.

The UAS-hid, Gal80ts parental stock used in our ablation experiments is obligately heterozygous over the CyO balancer (which does not have any other associated markers). For *Figures 3 and 6*, F1 progeny were inferred to be Hid+ and CyO- based on the lack of dbd neurons labeled with myr::HA, indicating a successful ablation. CyO by itself should not reduce the percentage of dbd labeling and thus show 100% dbd survival. The samples with ablated dbds were included in our analyses for *Figures 3 and 6*. Samples which retained 100% of dbds (presumably because they were Hid- and CyO+) were excluded.

In the critical period experiment, we did a 'No Ablation' control in which Hid, Gal80$^{ts}$ animals were reared alongside LacZ controls at 18°C. Hid, Gal80$^{ts}$ animals were identified based on the complete lack of dbd labeling, caused by the Gal80-inhibited expression of the UAS-driven dbd membrane label. All other animals in this group had 100% of dbds labeled and were excluded for likely being CyO+.

## Figure preparation

Micrographs in figures were prepared as either 3D projections in Imaris 9.5.1 (Bitplane AG, Zurich, Switzerland) or maximum intensity projections in FIJI (ImageJ 1.50d, https://imagej.net/Fiji). Scale bars are given for reference on maximum intensity projections but do not necessarily represent actual distances, as the tissue samples undergo changes in size during the tissue clearing protocol. For images exported from the Imaris software, the scale bars are assigned to match the scale at the 'center' of the 3D projection and therefore only serve as approximations of linear distance shown in the image. Pixel brightness was adjusted in some images for better visualization; all adjustments were made uniformly over the entire image, and uniformly across corresponding control and experimental images. Larval crawling traces are temporal projections made in FIJI (Image > Hyperstacks > Temporal Color-Code).

## Larval behavior assays

### TNT experiments

Newly hatched larvae were aged for 24 hr on standard cornmeal fly food at 25°C. At this time, five larvae were gently transferred to a 4×4 cm 1.2% agarose (Sigma, Lot# SLCD4639, Cat. A9539-500G) arena. Larvae were spaced apart to prevent collisions during recording. Prior to recording, larvae were acclimated to the arena for 2 min. The ambient temperature during recording was 20–22°C. Videos of individual larvae were collected at 5 frames/s for 1 min.

### Shibire experiments

Newly hatched larvae were aged for 22 hr on standard cornmeal fly food at 30°C. At this time, individual larvae were transferred to a 0.5 mm thick 1.2% agarose pad positioned on top of a 22×40 mm coverslip (Corning, Lot# 14418013, Cat. 2980–224, #1.5 thickness). The larva and coverslip were placed on top of a CherryTemp microfluidic chip (Cherry Biotech, Montreuil, France) with an 18°C surface temperature. Larvae were acclimated to the surface for 2 min prior to recording. Videos of individual larvae were collected at 5 frames/s for 1 min. For all behavior experiments, animals that failed to move during the 1 min recording were excluded from analysis.

### Chrimson experiments

Embryos and newly hatched larvae were aged for 23 hr under white light (~30,000 lux; larvae age 1.03× faster 28°C) with continuous access to +ATR yeast paste. At this time, a larva was gently transferred to a 4×4 cm 1.2% agarose (Sigma, Lot# SLCD4639, Cat. A9539-500G) arena, illuminated with white light (~30,000 lux). Prior to recording, larvae were acclimated to the arena for 2 min. The ambient temperature during recording was 20–22°C. Videos of individual larvae were collected for ~1 min and total number of locomotor waves was divided by total length of video recording (waves/s).

### Analysis of larval locomotor behavior

Movie processing and analysis was performed using FIJI (ImageJ 1.50d, https://imagej.net/Fiji). Locomotor waves were manually quantified. Forward and backward waves were summed together for

each animal. The number of waves for each animal was normalized to the control average of each independent experiment.

### Identification of top neuron partners in the TEM connectome

The L1 connectome of the larval *D. melanogaster* CNS was used (*Ohyama et al., 2015*). In the Connectivity widget of CATMAID, we obtained all postsynaptic partners for 'dbd_a1l', 'dbd_a1r', and all presynaptic partners for 'A08a_a1l' and 'A08a_a1r'. Partner neurons were filtered out if they were only present on one side of the brain, or if the sum of their synapses on left and right sides of the brain was fewer than 3. For the remaining neurons, we ranked the top 10 synaptically connected neurons downstream of dbd or upstream of A08a. If a neuron was present across multiple brain segments, we summed their synapses together and indicated this in the table where relevant. Neurotransmitter identities for some neurons were previously published in the literature and were included where possible.

### Sample definition and in-laboratory replication

A minimum of two independent experiments were done for each experiment. Sample sizes were determined based on previously published standards. Data points in this paper describe biological replicates (multiple individual animals of a corresponding genotype), as technical replicates (repeated measures of the same sample) were not relevant.

### Statistical analyses

Statistics were computed using Excel or Python (scipy.stats). All statistical tests used are listed in the figure legends. For comparisons between cumulative distributions in *Figure 2D*, a two-sample Kolmogorov-Smirnonv test was used as it is more appropriate for detecting differences across two cumulative distributions, rather than just looking for differences between the means of two groups. For remaining statistical tests, a Student's two-way t-test was used to detect differences in the mean distribution. An assumption of unequal variance was used, as the sample sizes were not sufficiently large to assume equal variance. Statistical outliers were maintained to show the variance in the data. p-Values are reported in the figures. n.s.=not significant, where $p > 0.05$. Plots were generated using Excel or Seaborn and Matplotlib packages in Python.

## Acknowledgements

We thank Dr Bruce Bowerman and Dr Molly Jud for the use of their CherryTemp system. We thank Alanna Sowles and Keiko Hirono for assistance with Imaris reconstructions. We thank Dr Sarah Ackerman, Dr Adam Miller, Dr Cris Niell, and Peter Newstein for constructive comments on the manuscript. Stocks obtained from the Bloomington Drosophila Stock Center (NIH P40OD018537) were used in this study.

## Additional information

### Competing interests

Chris Q Doe: Reviewing editor, *eLife*. The other author declares that no competing interests exist.

### Funding

| Funder | Grant reference number | Author |
|---|---|---|
| Howard Hughes Medical Institute | | Emily L Heckman Chris Q Doe |
| Eunice Kennedy Shriver National Institute of Child Health and Human Development | HD27056 | Emily L Heckman Chris Q Doe |

The funders had no role in study design, data collection and interpretation, or the decision to submit the work for publication.

## Author contributions
Emily L Heckman, Conceptualization, Resources, Data curation, Formal analysis, Validation, Investigation, Visualization, Methodology, Writing - original draft, Writing - review and editing; Chris Q Doe, Conceptualization, Data curation, Supervision, Funding acquisition, Writing - original draft, Project administration, Writing - review and editing

## Author ORCIDs
Emily L Heckman ![ORCID] http://orcid.org/0000-0002-0012-3364
Chris Q Doe ![ORCID] http://orcid.org/0000-0001-5980-8029

## Decision letter and Author response
Decision letter https://doi.org/10.7554/eLife.82093.sa1
Author response https://doi.org/10.7554/eLife.82093.sa2

---

# Additional files

## Supplementary files
• MDAR checklist

## Data availability
All data is included in the manuscript.

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
