## [Editor Report]

The findings reported in this article build nicely on previous work regarding the specification and growth of post-synaptic dendrites. Here the authors conduct an elegant genetic and anatomical analysis defining two opposing mechanisms that regulate post-synaptic dendrite morphogenesis and/or stabilization: presynaptic contact and neuronal activity. The data are of uniformly high quality, support the authors' major conclusions and offer important new insights into this fundamental aspect of circuit development.

---

## [Decision Letter]

**Decision letter after peer review:**

Thank you for submitting your article "Presynaptic contact and activity opposingly regulate postsynaptic dendrite outgrowth" for consideration by *eLife*. Your article has been reviewed by 3 peer reviewers, and the evaluation has been overseen by a Reviewing Editor and K VijayRaghavan as the Senior Editor. The following individual involved in review of your submission has agreed to reveal their identity: Greg J Bashaw (Reviewer #2).

Essential revisions:

1. Sample sizes for the cell ablations should be increased to provide statistical confidence in the robustness of the result. Currently, samples are 3-4.

2. Ackerman (Ackerman et al., 2021) posits a critical period for motorneuron dendrites between 0-8hrs after larval hatching. Here, this study suggests that UAS-hid activation following release of a tub-Gal80-ts blockade on Gal4 at the point of hatching might lead to changes on the postsynaptic dendrites of A08a during the same critical period. That might be so, but it will be important to have some more information on the ablation efficacy – e.g. how soon after releasing the Gal80 block on Gal4 is it evident, in the dbd cell body and at its axonal terminals?

3. Potential ambiguity regarding genotyping should be resolved, clearly. I assume that they did use marked balancers, but that some of the immuno was still uncertain (this can happen). Nevertheless, in combination with the absence of documentation about the timing and extent of cell death in the 'critical period' experiments, it seems warranted to seek clarification.

4. What is the biological context of dbd-A08a, eg with regard to developmental sequence of cell-cell interactions, connectivity, synaptic transmission or functional relevance to animal behaviour? Information on any of the above could be helpful to begin to interpret findings much better. Some data are presented on cell-cell interactions in Figure 1. If elaborated and paired with information on eventual connectivity, those might give sufficient context with which to interpret experimental findings.

5. Provide more context on the Dbd-A08a connection in the context of other inputs from the EM connectome. The Doe lab has contributed to analysis of the larval VNC connectome and published several papers on aspects of that. Thus, providing connectome context should not be a large burden. Specifically, this study prides itself, quite rightly, on the precision afforded by the genetic tools available. This is at odds with the complete lack of context offered to the reader on what is now known about dbd and A08a connectivity, which I would argue is crucial for the interpretation of experimental observations. For example, what are the downstream partners of the dbd proprioceptor, and where does A08a sit relative to others, both numerically with regard to proportion of synaptic connections, as well as functionally, where known? The same applies for the converse, from perspective of the A08a interneuron.

6. Being unfamiliar with the wiring diagram of the A08a interneuron and its inputs within the larval ventral nerve cord, I have a hard time in conceptualizing potential mechanisms of this plasticity between the two connected neurons they characterize. Can the authors comment more on what the EM reconstruction of A08a has shown in terms of input. Is the Dbd axonal input the primary one that normally synapses onto the medial dendrite branch of A08a, or is it one of only 10-20 inputs. If there are many others, what fraction of the synapses overall on the medial dendrite does Dbd make. It would dramatically change my interpretation of what is happening if Dbd only forms 1% of the synapses onto this A08a medial branch versus 50% of the inputs.

7. Greater attention should be given regarding attribution to prior work on this topic, particularly with respect to older work in the insect (not only *Drosophila*). There is a rich literature to which this paper adds. Placing this work in the proper context is essential.

*Reviewer #1 (Recommendations for the authors):*

1. Being unfamiliar with the wiring diagram of the A08a interneuron and its inputs within the larval ventral nerve cord, I have a hard time in conceptualizing potential mechanisms of this plasticity between the two connected neurons they characterize. Can the authors comment more on what the EM reconstruction of A08a has shown in terms of input. Is the Dbd axonal input the primary one that normally synapses onto the medial dendrite branch of A08a, or is it one of only 10-20 inputs. If there are many others, what fraction of the synapses overall on the medial dendrite does Dbd make. It would dramatically change my interpretation of what is happening if Dbd only forms 1% of the synapses onto this A08a medial branch versus 50% of the inputs. Similarly, how many neurons synapse onto A08a on all its dendritic branches. I would have a very different interpretation of what the activity manipulation data are suggesting if Dbd makes up a major component of A08a's synaptic input, versus only a small fraction. It would help the reader think about why Dbd plays such a prominent role in setting the overall dendritic field size of A08a if we can better conceptualize how much of A08a input and drive are likely coming from this one sensory neuron.

2. The authors use dendritic volume and branching of A08a as their primary output measurements for manipulations, which is fine. However, it would be nice to determine if synapse number is changing as well. I assume synapse number would scale to dendrite size, but using a tagged ACH receptor expressed in A08a could really show this important point (I assume this is a cholinergic neuron, but if not, that would be worth discussing as well, and other tagged receptor lines might be available for measuring postsynaptic site number). The Doe lab performed a similar analysis for their motor neuron dendrite plasticity work, so it was a bit unfortunate not to have a synaptic count for this Dbd-A08a dendrite plasticity work. It's not an essential addition, but would allow the authors to say that the dendrite scaling with Dbd activity is likely to be influencing synapse number as well.

3. Any thoughts on the postsynaptic signaling pathway in A08a that extends from postsynaptic activity to dendrite growth control. Would be worth a few sentences in the discussion to speculate on potential pathways. Is A08a a cholinergic neuron? Do the authors think this is likely through activity-dependent transcriptional changes in A08a, or rather local signaling events that enhance or restrict dendrite expansion through cytoskeletal regulation?

*Reviewer #2 (Recommendations for the authors):*

I believe I have touched on the most important points in the public review. There are a few areas where some additional clarification and/or documentation of the experiments may be helpful.

While the authors make a compelling argument for why dbd ablation does not lead to the expected changes in A08 dendrite elaboration (i.e. a balance between the loss of activity dependent suppression of growth and the loss of contact-dependent increase in growth), it is less clear to me why the experiments that induce neural activity only effect medial, but nod lateral dendrites. According to the simplest interpretation of the model, both dendritic domains should be affected. Some additional discussion here could be helpful

For the critical period data, it would be nice to comment/show the time difference in neuronal loss. Is it clear that DBD is effectively ablated at all time points?

*Reviewer #3 (Recommendations for the authors):*

Connectomic context: This study prides itself, quite rightly, on the precision afforded by the genetic tools available. This is at odds with the complete lack of context offered to the reader on what is now known about dbd and A08a connectivity, which I would argue is crucial for the interpretation of experimental observations. For example, what are the downstream partners of the dbd proprioceptor, and where does A08a sit relative to others, both numerically with regard to proportion of synaptic connections, as well as functionally, where known? The same applies for the converse, from perspective of the A08a interneuron.

Data interpretation: I may have missed it, but was left puzzled by two observations that I felt did not receive the attention deserved or suggested an interpretation that seemed inconguous: (a) that ablation of dbd caused A08a (lateral) dendrites to expand that normally do not make contact with dbd – why? (B) The suggestion in lines 276ff that silencing of dbd activity, which causes a more general dendritic overgrowth phenotype (i.e. not just the lateral A08a dendrites), demonstrated a suppression of dbd on dendritic growth. If indeed simply so, then why does the dbd ablation experiment lead to a different A08a dendritic growth phenotype?

On cell ablation: These experiments are suggestive, but really need careful controls that detail efficacy and timing of dbd killing. It is concerning that the authors were unclear as to the genotype of animals in which dbd neurons had not been ablated (definitely need to use fluorescently marked balancers). Caspase induced cell death can play out in different ways over time and space of a neurons, which may or may not have a significant impact on events and eventual phenotype interpretation. For example, what is the efficacy of dbd ablation prior to dbd axons reaching the CNS, during CNS ingrowth or after establishing contact with neurons such as A08a? What happens to growth cones and axons of dbd following hid mis-expression? Could remnants be signalling to A08a in some form, e.g. releasing signals such as ROS that are said to be upregulated with dying cells?

How do these events play out when hid expression is acutely induced, so as to establish if there is a critical period? Here, sample sizes are very small and in need of methodologically independent verification (ideally). One cannot necessarily extrapolate from using just the Gal4 line versus using a temperature sensitive Gal80 combination, which is likely dampening both dynamics and amplitude of Gal4 activity.

Language and reference to earlier work: This is a small but not insignificant comment, which may well be a personal perspective. To me it is epitomised by the statement in the discussion, line 256-257: "Interestingly, when paired with a pharmacological silencing manipulation, suprainnervation of the M-cell still resulted in elongated dendrites, suggesting that contact-based cues are sufficient to drive local dendrite outgrowth (Goodman and Model, 1990), matching our results." It is good to see reference to earlier work, yet the sentence concludes with "matching our results", i.e. their work matches 'ours', which is precisely the wrong way round. Correct would be to state that the findings of this new study compares well with earlier discoveries by others.

Moreover, given that this study tries to use a critical period as a 'selling point', I was surprised that significant earlier studies by a number of labs, e.g. Bate, Baines, Nose in particular had not be cited in any form, nor those of others, around critical periods in the adult, e.g. Broadie, Ramaswami, etc. This is poor scholarship at best, but potentially misleading. With scientific discovery it is helpful for readers to gain an awareness of how ideas emerge, develop and change, as is facilitated by building on and acknowledging appropriately the work of others.

Mechanistically, this work does not illuminate new ground, though could form the beginning of a study to also explore new understanding.

Therefore, in summary, I would gauge this manuscript not appropriate for publication in *eLife* as presented, though it would sit well with a number of alternative journals.

---

## [Author Response]

Essential revisions:1. Sample sizes for the cell ablations should be increased to provide statistical confidence in the robustness of the result. Currently, samples are 3-4.

This is a great point. We have increased our sample sizes so that each group now has 4-11 animals represented. Our conclusions remain the same: that only after ablating dbd at 0hrs alh there is a significant increase in A08a lateral dendrite length.

2. Ackerman (Ackerman et al., 2021) posits a critical period for motorneuron dendrites between 0-8hrs after larval hatching. Here, this study suggests that UAS-hid activation following release of a tub-Gal80-ts blockade on Gal4 at the point of hatching might lead to changes on the postsynaptic dendrites of A08a during the same critical period. That might be so, but it will be important to have some more information on the ablation efficacy – e.g. how soon after releasing the Gal80 block on Gal4 is it evident, in the dbd cell body and at its axonal terminals?

Constitutive expression of hid in embryonic dbd neurons on average results in ablation of 53% of dbd neurons within 9 hours (animals raised at 30***°***C, hid expression initiating at stage 14 with the onset of dbd-Gal4; ablation complete by stage 17; see figure S3). In these embryos, the entire dbd neuron (cell body to axon) is gone. When inactivation of Gal80-ts is used to express Hid at 48h after larval hatching (ALH), we observe ablation of 65% of dbd neurons 24h later (see figure 6). Thus, Hid-induced ablation of dbd can occur within hours of Hid expression. We clarify this in new text added to the legend of figure 6 and in the discussion.

3. Potential ambiguity regarding genotyping should be resolved, clearly. I assume that they did use marked balancers, but that some of the immuno was still uncertain (this can happen). Nevertheless, in combination with the absence of documentation about the timing and extent of cell death in the 'critical period' experiments, it seems warranted to seek clarification.

Great point. We have added a section to the Methods entitled, “Validation of Hid function and criteria for sample inclusion.” We hope this will clarify the sample inclusion criteria for Hid experiments and eliminate the genotype ambiguity. It says, "The UAS-hid, Gal80ts parental stock used in our ablation experiments is obligately heterozygous over the CyO balancer (which does not have any other associated markers). For Figures 3 and 6, F1 progeny were inferred to be Hid+ and CyO- based on the lack of dbd neurons labeled with myr::HA, indicating a successful ablation. CyO by itself should not reduce the percentage of dbd labeling and thus show 100% dbd survival. The samples with ablated dbd’s were included in our analyses for Figures 3 and 6. Samples which retained 100% of dbd’s (presumably because they were Hid- and CyO+) were excluded."

4. What is the biological context of dbd-A08a, eg with regard to developmental sequence of cell-cell interactions, connectivity, synaptic transmission or functional relevance to animal behaviour? Information on any of the above could be helpful to begin to interpret findings much better. Some data are presented on cell-cell interactions in Figure 1. If elaborated and paired with information on eventual connectivity, those might give sufficient context with which to interpret experimental findings.

For dbd neurons, we show the timeline of the dbd driver line in figure S1; its expression is first detectable in embryonic stage 14, when dbd axons are immature growth cones and are at various stages of growth into the neuropil. For A08a neurons, we add new data to figure 1 showing that the A08a driver line is first expressed at 2h ALH, at the time dendrites have already formed (new figure 1E,F). The timing of A08a driver expression precludes us from knowing the developmental status of A08a dendrites at the time of first contact with dbd, which we expand upon in the Discussion. We thank the reviewer for helping us clarify the timing of dbd and A08a driver lines.

Regarding connectivity of the two neurons, please see the next comment/response (#5). Regarding function, several labs have shown that dbd is a proprioceptor, whose activity during larval crawling is co-incident with body segment relaxation (Suslak et al., 2015; Vaadia et al., 2019); the in vivo role of A08a during behavior is unknown, although it’s participation in a central pattern generating network (Itakura et al., 2015) is consistent with a role in regulating larval crawling. We have this added information on dbd and A08a function to the Introduction.

5. Provide more context on the Dbd-A08a connection in the context of other inputs from the EM connectome. The Doe lab has contributed to analysis of the larval VNC connectome and published several papers on aspects of that. Thus, providing connectome context should not be a large burden. Specifically, this study prides itself, quite rightly, on the precision afforded by the genetic tools available. This is at odds with the complete lack of context offered to the reader on what is now known about dbd and A08a connectivity, which I would argue is crucial for the interpretation of experimental observations. For example, what are the downstream partners of the dbd proprioceptor, and where does A08a sit relative to others, both numerically with regard to proportion of synaptic connections, as well as functionally, where known? The same applies for the converse, from perspective of the A08a interneuron.

Some of the connectivity information is present in the Sales et al., 2019 paper which is the foundation for this research advance, but your point is well taken -- most readers will read this research advance in the absence of the initial paper, and thus it is important to add and expand upon the previous connectivity data in this manuscript. We have added a new Table 1 that lists the top 10 strongest dbd target neurons (by synapse number) and found that A08a is #6. We added a new Table 2, that lists the top 10 strongest inputs into A08a, and found that dbd neurons are #1. Table 2 also indicates the specific axon/dendrite domain of A08a targeted by each input neuron; each input neuron selectively synapses with a specific A08a subcellular domain: medial dendrites, lateral dendrites, or output/axon domain. Table 2 also describes the neurotransmitter used by each input into A08a (where known). Finally, we have added a new figure S1 showing TEM reconstructions of each top 10 input neurons for A08a.

6. Being unfamiliar with the wiring diagram of the A08a interneuron and its inputs within the larval ventral nerve cord, I have a hard time in conceptualizing potential mechanisms of this plasticity between the two connected neurons they characterize. Can the authors comment more on what the EM reconstruction of A08a has shown in terms of input. Is the Dbd axonal input the primary one that normally synapses onto the medial dendrite branch of A08a, or is it one of only 10-20 inputs. If there are many others, what fraction of the synapses overall on the medial dendrite does Dbd make. It would dramatically change my interpretation of what is happening if Dbd only forms 1% of the synapses onto this A08a medial branch versus 50% of the inputs.

As stated above, we appreciate the opportunity to provide readers with updated information from the L1 connectome in the form of two new tables listing the top 10 postsynaptic partners of dbd (Table 1) and the top 10 presynaptic partners of A08a (Table 2). Based on synapse counts, dbd is the input to A08a overall, and is also the only excitatory neuron in the top 10 to synapse with the A08a medial dendrite. Among the top 10 neurons in Table 2, dbd synapses make up 52.4% of the total synapses targeted to the A08a medial dendrite. We hope this information adds the appropriate context for how a single neuron population can exert such an effect on the dendritic architecture of A08a. Furthermore, we have added a new figure S1 showing TEM reconstructions of each top 10 input neurons for A08a.

7. Greater attention should be given regarding attribution to prior work on this topic, particularly with respect to older work in the insect (not only *Drosophila*). There is a rich literature to which this paper adds. Placing this work in the proper context is essential.

We are sorry to have left out these relevant citations, especially the embryonic critical period work of Giachello and Baines. We have expanded our discussion to include numerous previously missing citations relevant to critical periods in *Drosophila* embryos and early larvae, as well as other insects (see below). We are sorry the reviewers had to call us out on this, but very glad they did.

Embryo/larval critical period

Jarecki / Keshishian 1995

Tripodi / Landgraf 2008

Hartwig / Ramaswami 2008

Crisp / Bate 2011

Fushiki / Nose 2013

Giachello / Baines 2015

Giachello / Baines 2021

Adult critical period

Doll / Broadie 2015

Doll / Broadie 2016

Doll / Broadie 2017

Golovin / Broadie 2019

Golovin / Broadie 2021

Manduca and honeybee critical periods

Caglayan / Gilbert 1987

Levine / Bate 1986

Morgan / Mercer 1998

Hartfelder / Hepperle 1995

Elekonich / Robinson 2003

We modify the discussion to say:

" Critical periods are developmental windows of heightened neural plasticity that are present from vertebrates (Kalb, 1994; Keck et al., 2017; Takesian and Hensch, 2013; Walton et al., 1992) to insects (Caglayan and Gilbert, 1987; Elekonich et al., 2003; Hartfelder et al., 1995; Levine et al., 1986; Morgan et al., 1998). In *Drosophila*, critical periods have been identified in the late embryo/early larva (Ackerman et al., 2021; Crisp et al., 2011; Fushiki et al., 2013; Giachello et al., 2021; Giachello and Baines, 2015; Hartwig et al., 2008; Jarecki and Keshishian, 1995; Tripodi et al., 2008) and in the newly-eclosed adult (Doll et al., 2017; Doll and Broadie, 2016, 2015; Golovin et al., 2021, 2019). Altered activity during the critical period can lead to long-lasting defects in neuronal morphology and behavior (Ackerman et al., 2021; Crisp et al., 2011; Fushiki et al., 2013; Giachello et al., 2021; Giachello and Baines, 2015; Hartwig et al., 2008; Jarecki and Keshishian, 1995; Tripodi et al., 2008). In the *Drosophila* embryonic motor system, dendrite length can be homeostatically modified by levels of activity (Tripodi et al., 2008). The motor dendrites lose the capacity to undergo activity-dependent remodeling at 8hrs alh, early in larval life. The closure of this critical period is governed by astrocytes – their infiltration into the neuropil coincides with critical period closure and their contact with motor dendrites prevents precocious dendrite extension/retraction (Ackerman et al., 2021; Stork et al., 2014)."

Reviewer #1 (Recommendations for the authors):1. Being unfamiliar with the wiring diagram of the A08a interneuron and its inputs within the larval ventral nerve cord, I have a hard time in conceptualizing potential mechanisms of this plasticity between the two connected neurons they characterize. Can the authors comment more on what the EM reconstruction of A08a has shown in terms of input. Is the Dbd axonal input the primary one that normally synapses onto the medial dendrite branch of A08a, or is it one of only 10-20 inputs. If there are many others, what fraction of the synapses overall on the medial dendrite does Dbd make. It would dramatically change my interpretation of what is happening if Dbd only forms 1% of the synapses onto this A08a medial branch versus 50% of the inputs. Similarly, how many neurons synapse onto A08a on all its dendritic branches. I would have a very different interpretation of what the activity manipulation data are suggesting if Dbd makes up a major component of A08a's synaptic input, versus only a small fraction. It would help the reader think about why Dbd plays such a prominent role in setting the overall dendritic field size of A08a if we can better conceptualize how much of A08a input and drive are likely coming from this one sensory neuron.

Some of the connectivity information is present in the Sales et al., 2019 paper which is the foundation for this research advance, but your point is well taken -- most readers will read this research advance paper in the absence of the initial paper, and thus it is important to add and expand the connectivity data in this manuscript. We have added a new Table 1, that lists the top 10 strongest (most synapses) dbd target neurons and found that A08a is #6. We added a new Table 2, that lists the top 10 strongest inputs into A08a, and found that dbd is #1. Among the top 10 neurons in Table 2, dbd synapses make up 52.4% of the total synapses targeted to the A08a medial dendrite. Table 2 also indicates the specific axon/dendrite domain of A08a targeted by each input neuron; each input neuron selectively synapses with a specific A08a arbor: medial dendrites, lateral dendrites, or output domain. Table 2 also describes the neurotransmitter used by each input into A08a (where known). Finally, we have added a new figure s1 showing TEM reconstructions of each top input neuron with A08a.

2. The authors use dendritic volume and branching of A08a as their primary output measurements for manipulations, which is fine. However, it would be nice to determine if synapse number is changing as well. I assume synapse number would scale to dendrite size, but using a tagged ACH receptor expressed in A08a could really show this important point (I assume this is a cholinergic neuron, but if not, that would be worth discussing as well, and other tagged receptor lines might be available for measuring postsynaptic site number). The Doe lab performed a similar analysis for their motor neuron dendrite plasticity work, so it was a bit unfortunate not to have a synaptic count for this Dbd-A08a dendrite plasticity work. It's not an essential addition, but would allow the authors to say that the dendrite scaling with Dbd activity is likely to be influencing synapse number as well.

We appreciate the statement that this experiment is "not an essential addition" because the first author has started her postdoc since submission of this manuscript back in July, plus it would take many months.

3. Any thoughts on the postsynaptic signaling pathway in A08a that extends from postsynaptic activity to dendrite growth control. Would be worth a few sentences in the discussion to speculate on potential pathways. Is A08a a cholinergic neuron? Do the authors think this is likely through activity-dependent transcriptional changes in A08a, or rather local signaling events that enhance or restrict dendrite expansion through cytoskeletal regulation?

Thanks for the opportunity to add some points of discussion here. There is a large body of work on the connection between activity and changes in dendritic arbor size. Many of the best characterized pathways linking activity to dendrite growth are calcium based, and can function globally by regulating gene expression (e.g. CREB, Wnt). We add the following new text to the Discussion to provide this context:

"How might activity limit dendrite growth throughout the postsynaptic neuron? Sustained alterations to neurotransmission would likely impact postsynaptic calcium levels and therefore intracellular calcium signaling. Indeed, studies in *Drosophila* show PKA regulation of dendritic arbor size (Baines, 2003; Tripodi et al., 2008). Calcium can also signal through regulators of transcription, allowing the effect of activity to potentially be exerted neuron-wide through changes in gene expression (Sun et al., 1994). While calcium signaling pathways have probably been the most explored among activity-dependent mechanisms regulating dendrite growth, it is likely one of many pathways that the cell could deploy. For example, Wnt signaling is also recruited to inhibit dendrite elongation when activity levels are high (Singh et al., 2010). In the future, it will be interesting to explore the context in which each type of signaling pathway is engaged, and how they could oppose the effect of growth-promoting contact."

Reviewer #2 (Recommendations for the authors):I believe I have touched on the most important points in the public review. There are a few areas where some additional clarification and/or documentation of the experiments may be helpful.While the authors make a compelling argument for why dbd ablation does not lead to the expected changes in A08 dendrite elaboration (i.e. a balance between the loss of activity dependent suppression of growth and the loss of contact-dependent increase in growth), it is less clear to me why the experiments that induce neural activity only effect medial, but nod lateral dendrites. According to the simplest interpretation of the model, both dendritic domains should be affected. Some additional discussion here could be helpful

We are sorry this was not clear. Figure 5 shows that activating dbd leads to a diminishing of the medial arbor, and a modest but statistically insignificant decrease in lateral arbor size. We’ve added some potential reasons for this to the Discussion. These include (i) the possibility that dbd is simultaneously providing feed-forward inhibition onto A08a via its connection with the glutamatergic A02d and (ii) that while CsChrimson has been used in long-term activation paradigms before, it is unknown whether the channel can actually remain open for the entire stimulus duration.

For the critical period data, it would be nice to comment/show the time difference in neuronal loss. Is it clear that DBD is effectively ablated at all time points?

We thank the reviewer for this comment, and have clarified this point in the text. Constitutive expression of Hid in embryonic dbd neurons results ablation of 53% of dbd neurons within 9 hours (animals raised at 30***°***C, Hid expression initiating at stage 14; ablation complete by stage 17 (see figure S3)). In larvae, Gal80-ts inactivation at 48h ALH to express Hid results in 65% dbd ablation within 24h, by 72h ALH (see figure 6A). Thus, Hid-induced ablation of dbd occurs by 24h, and probably much sooner. We clarify this in new text added to the discussion (lines 237-241).

Reviewer #3 (Recommendations for the authors):Connectomic context: This study prides itself, quite rightly, on the precision afforded by the genetic tools available. This is at odds with the complete lack of context offered to the reader on what is now known about dbd and A08a connectivity, which I would argue is crucial for the interpretation of experimental observations. For example, what are the downstream partners of the dbd proprioceptor, and where does A08a sit relative to others, both numerically with regard to proportion of synaptic connections, as well as functionally, where known? The same applies for the converse, from perspective of the A08a interneuron.

Some of the connectivity information is present in the Sales et al., 2019 paper which is the foundation for this research advance, but your point is well taken -- most readers will read this research advance paper in the absence of the initial paper, and thus it is important to add and expand the connectivity data in this manuscript. We have added a new Table 1, that lists the top 10 strongest dbd target neurons (by synapse number) and found that A08a is #6. We added a new Table 2, that lists the top 10 strongest inputs into A08a, and found that dbd’s are #1. Table 2 also indicates the specific axon/dendrite domain of A08a targeted by each input neuron; each input neuron selectively synapses with a specific A08a subcellular domain: medial dendrites, lateral dendrites, or axon/output domain. Table 2 also describes the neurotransmitter used by each input into A08a (where known). This shows that dbd is also the top excitatory input to A08a, and the only excitatory input to the medial arbor among the top 10 upstream partners. Finally, we have also added a new figure S1 showing TEM reconstructions of each top input neuron with A08a.

Data interpretation: I may have missed it, but was left puzzled by two observations that I felt did not receive the attention deserved or suggested an interpretation that seemed inconguous: (a) that ablation of dbd caused A08a (lateral) dendrites to expand that normally do not make contact with dbd – why? (B) The suggestion in lines 276ff that silencing of dbd activity, which causes a more general dendritic overgrowth phenotype (i.e. not just the lateral A08a dendrites), demonstrated a suppression of dbd on dendritic growth. If indeed simply so, then why does the dbd ablation experiment lead to a different A08a dendritic growth phenotype?

Regarding question (a), we hypothesize that ablation of dbd causes loss of A08a excitation, resulting in a homeostatic response to increase all A08a dendrites, both medial and lateral; enlargement of A08a medial dendrites is masked, however, by the loss of contact-based dendrite outgrowth. Thus, the lateral dendrites experience only homeostatic growth, whereas medial dendrites experience both homeostatic growth and loss of contact-induced dendrite growth which cancel each other out. Regarding question (b), the same explanation (or hypothesis) holds for this question as well, which results in the differential effect on the lateral arbor (only homeostatic growth inputs) versus medial arbor (both homeostatic growth inputs AND reduced dendrite growth due to loss of contact-based growth cues). We have revised the Discussion to be clearer on these points.

Or we can quote reviewer 2 for their elegant phrasing: "… the authors make a compelling argument for why dbd ablation does not lead to the expected changes in A08 dendrite elaboration (i.e. a balance between the loss of activity dependent suppression of growth and the loss of contact-dependent increase in growth)."

On cell ablation: These experiments are suggestive, but really need careful controls that detail efficacy and timing of dbd killing. It is concerning that the authors were unclear as to the genotype of animals in which dbd neurons had not been ablated (definitely need to use fluorescently marked balancers).

Great points, thanks for prompting us to clarify these important aspects of our work.

Regarding the timing of dbd ablation. We have clarified this point in the text. Constitutive expression of Hid in embryonic dbd neurons results ablation of 53% of dbd neurons within 9 hours (animals raised at 30***°***C, Hid expression initiating at stage 14; ablation complete by stage 17 (see figure S3)). In larvae, Gal80-ts inactivation at 48h ALH to express Hid results in 65% dbd ablation within 24h, by 72h ALH (see figure 6). Thus, Hid-induced ablation of dbd occurs by 24h, and probably much sooner. We clarify this in new text added to the discussion (lines 237-241).

Regarding the genotype identification. We have added a section to the Methods entitled, “Validation of Hid function and criteria for sample inclusion.” We hope this will eliminate the genotype ambiguity. It says, "The UAS-hid, Gal80ts parental stock used in our ablation experiments is obligately heterozygous over the CyO balancer (which does not have any other associated markers). For Figures 3 and 6, F1 progeny were inferred to be Hid+ and CyO- based on the lack of dbd neurons labeled with myr::HA, indicating a successful ablation. CyO by itself should not reduce the percentage of dbd labeling and thus show 100% dbd survival. The samples with ablated dbd’s were included in our analyses for Figures 3 and 6. Samples which retained 100% of dbd’s (presumably because they were Hid- and CyO+) were excluded."

Caspase induced cell death can play out in different ways over time and space of a neurons, which may or may not have a significant impact on events and eventual phenotype interpretation. For example, what is the efficacy of dbd ablation prior to dbd axons reaching the CNS, during CNS ingrowth or after establishing contact with neurons such as A08a? What happens to growth cones and axons of dbd following hid mis-expression? Could remnants be signalling to A08a in some form, e.g. releasing signals such as ROS that are said to be upregulated with dying cells?

We have clarified this point in the text. Constitutive expression of Hid in embryonic dbd neurons results in ablation of 53% of dbd neurons within 9 hours (animals raised at 30***°***C, Hid expression initiating at stage 14; ablation complete by stage 17; see figure S3). This means for our experiments in Figure 3, ablation is occurring with a similar timeline as dbd synapse formation.

In Hid brains few, if any, dbd fragments remain; it seems likely that they are cleared by macrophages, as has been shown for other cellular debris. We are confident that dbd activity does not persist at normal levels after ablation (e.g. via signaling from cell fragments) because we observe a phenotype in A08a following dbd ablation that is similar to silencing of dbd. The reviewer suggests that ROS signaling from dying dbd’s could have contributed to the expansion of A08a dendrites, but we think this is probably not the case; it’s been elegantly shown in *Drosophila* motor neurons that ROS signaling induces an overactivation phenotype in these dendrites (i.e. diminishment) (Oswald et al., 2018).

Regarding the timing of dbd ablation following Gal80-ts heat inactivation, we find that in larvae, Gal80-ts inactivation at 48h ALH results in 65% dbd ablation within 24h (i.e. by 72h ALH; see figure 6). Thus, in this example Hid-induced ablation of dbd occurs within 24h, and probably much sooner. We clarify this in new text added to the discussion (lines 237-241).

How do these events play out when hid expression is acutely induced, so as to establish if there is a critical period? Here, sample sizes are very small and in need of methodologically independent verification (ideally). One cannot necessarily extrapolate from using just the Gal4 line versus using a temperature sensitive Gal80 combination, which is likely dampening both dynamics and amplitude of Gal4 activity.

We agree with this view. We have changed the text to replace "matching our results" with "our results are consistent with these findings".

Language and reference to earlier work: This is a small but not insignificant comment, which may well be a personal perspective. To me it is epitomised by the statement in the discussion, line 256-257: "Interestingly, when paired with a pharmacological silencing manipulation, suprainnervation of the M-cell still resulted in elongated dendrites, suggesting that contact-based cues are sufficient to drive local dendrite outgrowth (Goodman and Model, 1990), matching our results." It is good to see reference to earlier work, yet the sentence concludes with "matching our results", i.e. their work matches 'ours', which is precisely the wrong way round. Correct would be to state that the findings of this new study compares well with earlier discoveries by others.Moreover, given that this study tries to use a critical period as a 'selling point', I was surprised that significant earlier studies by a number of labs, e.g. Bate, Baines, Nose in particular had not be cited in any form, nor those of others, around critical periods in the adult, e.g. Broadie, Ramaswami, etc. This is poor scholarship at best, but potentially misleading. With scientific discovery it is helpful for readers to gain an awareness of how ideas emerge, develop and change, as is facilitated by building on and acknowledging appropriately the work of others.

We are sorry to have left out these relevant citations, especially the embryonic critical period work of Giachello and Baines. We have expanded our discussion to include numerous previously missing citations relevant to critical periods in *Drosophila* embryos and early larvae, as well as other insects (see below). We are sorry the reviewers had to call us out on this, but very glad they did.

Embryo/larval critical period

Jarecki / Keshishian 1995

Tripodi / Landgraf 2008

Hartwig / Ramaswami 2008

Crisp / Bate 2011

Fushiki / Nose 2013

Giachello / Baines 2015

Giachello / Baines 2021

Adult critical period

Doll / Broadie 2015

Doll / Broadie 2016

Doll / Broadie 2017

Golovin / Broadie 2019

Golovin / Broadie 2021

Manduca and honeybee critical periods

Caglayan / Gilbert 1987

Levine / Bate 1986

Morgan / Mercer 1998

Hartfelder / Hepperle 1995

Elekonich / Robinson 2003

We modify the discussion to say:

" Critical periods are developmental windows of heightened neural plasticity that are present from vertebrates (Kalb, 1994; Keck et al., 2017; Takesian and Hensch, 2013; Walton et al., 1992) to insects (Caglayan and Gilbert, 1987; Elekonich et al., 2003; Hartfelder et al., 1995; Levine et al., 1986; Morgan et al., 1998). In *Drosophila*, critical periods have been identified in the late embryo/early larva (Ackerman et al., 2021; Crisp et al., 2011; Fushiki et al., 2013; Giachello et al., 2021; Giachello and Baines, 2015; Hartwig et al., 2008; Jarecki and Keshishian, 1995; Tripodi et al., 2008) and in the newly-eclosed adult (Doll et al., 2017; Doll and Broadie, 2016, 2015; Golovin et al., 2021, 2019). Altered activity during the critical period can lead to long-lasting defects in neuronal morphology and behavior (Ackerman et al., 2021; Crisp et al., 2011; Fushiki et al., 2013; Giachello et al., 2021; Giachello and Baines, 2015; Hartwig et al., 2008; Jarecki and Keshishian, 1995; Tripodi et al., 2008). In the *Drosophila* embryonic motor system, dendrite length can be homeostatically modified by levels of activity (Tripodi et al., 2008). The motor dendrites lose the capacity to undergo activity-dependent remodeling at 8hrs alh, early in larval life. The closure of this critical period is governed by astrocytes – their infiltration into the neuropil coincides with critical period closure and their contact with motor dendrites prevents precocious dendrite extension/retraction (Ackerman et al., 2021; Stork et al., 2014)."